# From the first touch to biofilm establishment by the human pathogen *Candida glabrata*: a genome-wide to nanoscale view

Mafalda Cavalheiro[1,2], Diana Pereira [1,2], Cécile Formosa-Dague[3], Carolina Leitão[1,2], Pedro Pais [1,2], Easter Ndlovu[4], Romeu Viana[1,2], Andreia I. Pimenta[1,2], Rui Santos[1,2], Azusa Takahashi-Nakaguchi[5], Michiyo Okamoto[5], Mihaela Ola[6], Hiroji Chibana[5], Arsénio M. Fialho[1,2], Geraldine Butler [6], Etienne Dague [4✉] & Miguel C. Teixeira [1,2✉]

*Candida glabrata* is an opportunistic pathogen that adheres to human epithelial mucosa and forms biofilm to cause persistent infections. In this work, Single-cell Force Spectroscopy (SCFS) was used to glimpse at the adhesive properties of *C. glabrata* as it interacts with clinically relevant surfaces, the first step towards biofilm formation. Following a genetic screening, RNA-sequencing revealed that half of the entire transcriptome of *C. glabrata* is remodeled upon biofilm formation, around 40% of which under the control of the transcription factors CgEfg1 and CgTec1. Using SCFS, it was possible to observe that CgEfg1, but not CgTec1, is necessary for the initial interaction of *C. glabrata* cells with both abiotic surfaces and epithelial cells, while both transcription factors orchestrate biofilm maturation. Overall, this study characterizes the network of transcription factors controlling massive transcriptional remodelling occurring from the initial cell-surface interaction to mature biofilm formation.

[1] Department of Bioengineering, Instituto Superior Técnico, Universidade de Lisboa, Lisbon, Portugal. [2] Biological Sciences Research Group, iBB - Institute for Bioengineering and Biosciences, Instituto Superior Técnico, Lisbon, Portugal. [3] TBI, Université de Toulouse, INSA, INRAE, CNRS, Toulouse, France. [4] LAAS-CNRS, Université de Toulouse, CNRS, Toulouse, France. [5] Medical Mycology Research Center (MMRC), Chiba University, Chiba, Japan. [6] School of Biomedical and Biomolecular Sciences, Conway Institute, University College Dublin, Dublin, Ireland. ✉email: edague@laas.fr; mnpct@tecnico.ulisboa.pt

The use of medical devices has increased over the years, driven by the necessity to improve the quality of life and by the fast development of new technology[1]. Although bringing enormous advantages, medical devices also have a dark side related to the emergence of microbial infections[1–3]. Medical devices, once inserted in the human host, provide a favorable surface in which microorganisms may develop biofilms, giving rise to persistent colonization and, often, a very hard to eradicate source of infection[4]. Once the biofilm is formed on such devices, disseminated cells can move to other host niches and just as well originate new biofilms[5].

*Candida glabrata* is an opportunistic pathogen that is able to use such surfaces to form biofilms and prevail in the human host[5–7]. It is considered the second most common cause of candidiasis[8, 9], being associated to both invasive disseminated infections and oral, esophageal, or vulvovaginal candidiasis[10–13]. *C. glabrata* is able to adhere to abiotic surfaces such as polyvinyl chloride, polyurethane, polystyrene, silicone, Teflon, and denture acrylic surfaces[7, 14–18], but it is as well able to adhere to different epithelial cells such as the human cultured epithelial cells HEp2[19] or the human vaginal epithelial cell line VK2/E6E7[20].

Adhesion is a complex interplay between physico-chemical interactions (hydrophobicity, electrostatic interactions) and adhesin-mediated interactions. Adhesins are glycosylphosphatidylinositol cell wall-anchored proteins, divided into seven subfamilies based on the phylogenetic analysis of their putative N-terminal ligand-binding regions[21, 22]. The most studied *C. glabrata* family of adhesins is the Epithelial Adhesin (EPA) family, given the importance of its 17 members in the adherence of this fungal pathogen[21]. The cellular proliferation and production of an extracellular matrix (ECM) gives continuation to *C. glabrata* biofilm formation, in a thick structure of yeast cells that exhibits higher resistance to antifungal drugs than planktonic cells[23, 24], being the perfect niche for the protection of *C. glabrata* cells.

In this work, *C. glabrata*'s ability to adhere to different clinically relevant surfaces was assessed by single-cell force spectroscopy (SCFS), revealing the forces established between *C. glabrata* and plastic surfaces used in medical devices, but also the ability to establish interactions with human vaginal epithelial cells (VK2/E6E7 cell line). From a screening of potential transcription factors involved in the control of biofilm formation, only two were found to be necessary for adhesion to human vaginal epithelial cells and biofilm formation in *C. glabrata*: CgEfg1 and CgTec1. Given their relevance on biofilm formation, the study of the transcriptomic remodeling occurring in *C. glabrata* cells from planktonic cultivation to 24 h of biofilm formation, in the presence or absence of

*CgEFG1* or *CgTEC1* genes, was pursuit through RNA-sequencing (RNA-seq). To assess the importance of CgEfg1 and CgTec1 at the scale of cell-surface molecular interactions, SCFS was used to measure how the expression of *CgEFG1* and *CgTEC1* modulate the interaction forces between *C. glabrata* and clinically related surfaces.

## Results

**C. glabrata is strongly adherent to plastic surfaces used in medical devices.** Given the importance of adhesion as the first step of biofilm formation, the interaction forces established between a single KUE100 *C. glabrata* cell and glass, polystyrene, silicone elastomer, and polyvinyl chloride were quantified. To do so, a noninvasive technique was used, SCFS, with tipless cantilevers coated with concanavalin A (conA) to immobilize a single *C. glabrata* cell. Each cell probe was slowly approached towards the surface material, for different periods of time: 0, 0.5, 1, and 5 s, and was withdrawn at constant speed. The cell probes and surfaces were kept submersed in acetate buffer, pH 5.2, for all the experiments. Two hundred and fifty-six force–distance curves were recorded according to a $10 \times 10\ \mu m^2$ force map, between the *C. glabrata* KUE100 single-cell probes and the different materials.

*C. glabrata* interaction with each surface was studied based on the maximum adhesion force, work of adhesion, and rupture distance measured on the force–distance curves, and was plotted as repartition histograms. The maximum adhesion force measured corresponds to the maximum adhesion force felt during the entire interaction, the work of adhesion consists in the complete set of forces established during the interaction (area under the retraction force curve) and the rupture distance is the distance from the contact point to the last force established during interaction. The mean value and their Standard Deviation (SD) were extracted from a gaussian fit.

Interaction forces between a yeast cell and glass show a maximum adhesion force of about 1 nN (Fig. 1a), work of adhesion of $6.58 \times 10^{-16}$ J (Fig. 1b), and rupture distance of ~1.2 μm (Fig. 1c). The representative force–distance curve for the interaction with glass is depicted in Fig. 1d (black line), showing a small peak of adhesion and an elongation before the rupture. The adhesion of *C. glabrata* to other hydrophobic surfaces was further tested, using currently used medical materials, namely polystyrene, silicone elastomer, and polyvinyl chloride[7, 14, 25]. The values of maximum adhesion force towards polystyrene, silicone elastomer, and polyvinyl chloride obtained were ~14, ~10, and ~19 nN, respectively. All of these *C. glabrata*–material interactions are significantly more adhesive than towards glass (Fig. 1a, b),

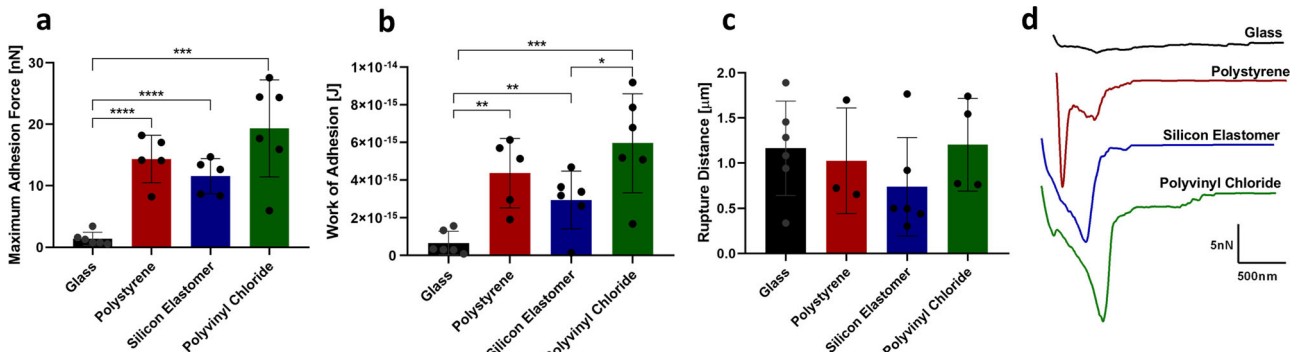

**Fig. 1 Interaction of *C. glabrata* wild-type strain KUE100 with glass, polystyrene, silicone elastomer, and polyvinyl chloride by SCFS.** Average of the: **a** maximal adhesion force, **b** work of adhesion, and **c** rupture distance measured on each retraction curve. **d** Representative force–distance curves of the interaction with glass (black), polystyrene (red), silicone elastomer (blue), and polyvinyl chloride (green). For every condition, at least 5 yeast cells, from at least 3 independent cell cultures, were immobilized on the cantilever for the interaction of each material and 256 force–distance curves were recorded. Error bars indicate SDs. *$P < 0.05$; **$P < 0.01$; ***$P < 0.001$; ****$P < 0.0001$, $n \geq 3$.

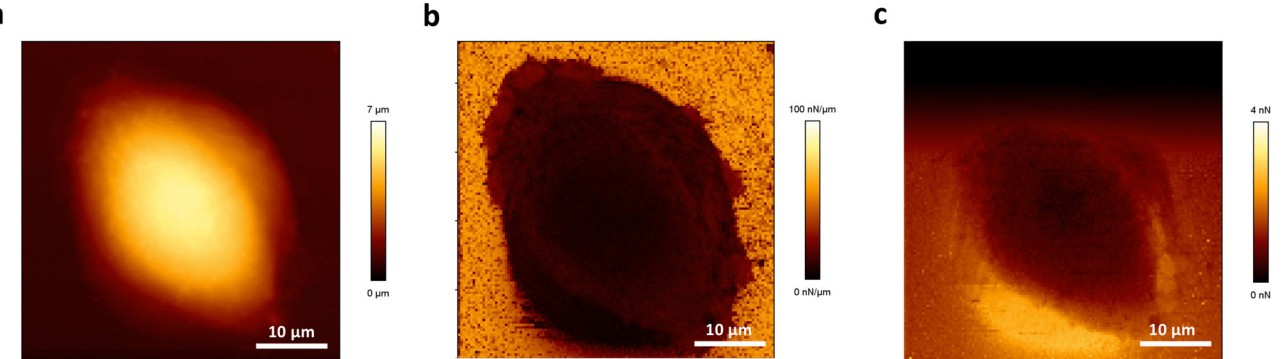

**Fig. 2 AFM imaging of human vaginal epithelial VK2/E6E7 cells in the QI^TM. a** Height image based on the contact point position, **b** spring constant map corresponding to the slope of the approach force-distance curves, and **c** adhesion map corresponding to the maximum adhesion force on the retraction force-distance curves.

although no differences are observed in terms of rupture distance (Fig. 1c). The representative force–distance curves show very high peaks of adhesion with small elongation before the rupture of the interaction, suggesting that mainly nonspecific interactions, such as hydrophobic or electrostatic[26, 27], are at play when a *C. glabrata* cell adheres to polystyrene, silicone elastomer, or polyvinyl chloride (Fig. 1d red, blue, and green, respectively). Moreover, an adhesion frequency of 100% was observed for *C. glabrata* interaction with all the abiotic surfaces tested.

Increasing the contact time of the yeast cell and surface from 0 to 0.5, 1, or 5 s is accompanied by an increase in maximum adhesion force and work of adhesion (Supplementary Fig. 1). For glass and polystyrene, *C. glabrata* adhesion is promoted by the increase of the contact time between yeast cell and material surface, suggesting sensitivity to contact. However, this behavior was not expected for the interaction with materials given that it is typical for biological interactions, mainly adhesin interactions, depending on the association rate[28]. For silicone elastomer and polyvinyl chloride, any of the contact times tested lead to higher adhesion force than 0 s of contact time, but no difference exists between 0.5, 1 or 5 s, suggesting that all the possible interactions are established in a short amount of time (<0.5 s) and are then unsensitive to contact time. Interestingly, the rupture distance does not change with any of the contact times used: 0, 0.5, 1, or 5 s. Nevertheless, it is clear that extending contact time strengthens the adhesion of this pathogenic yeast to the medical materials tested.

**C. glabrata adheres to human vaginal epithelial cells**. To study the interaction of the wild-type *C. glabrata* KUE100 with human epithelial cells, the human vaginal epithelial cell line VK2/E6E7 was selected due to the frequent vulvovaginal infections *C. glabrata* is known to cause[29]. The epithelial cells were first characterized by imaging them using atomic force microscopy (AFM), in the Quantitative Imaging^TM (QI^TM) mode (Fig. 2)[30, 31]. One very obvious characteristic of these cells is the height of the nucleus, more than 10 μm on almost all tested cells, which makes AFM imaging challenging. Comparing Fig. 2 to the QI^TM images of other living adherent mammalians cells, such as Chinese hamster ovary and human colorectal tumor cells, it is possible to observe similarities in height[30].

The interactions established between a single *C. glabrata* cell and a single human vaginal epithelial cell were studied resorting to SCFS. *C. glabrata* cell probes were prepared with tipless cantilevers modified with conA. The epithelial cells were grown in Keratinocyte Serum-Free (KSF) medium, at 37 °C, 5% $CO_2$, and were maintained in these conditions during the experiments. Each cell probe was approached towards a given epithelial

cell with an extended speed of 20 μm/s, for different periods of time—5, 10, 30, and 60 s—and was withdrawn at constant speed. Adhesion maps were recorded during constant approach and retraction of the *C. glabrata* KUE100 single-cell probes towards each single epithelial cell.

*C. glabrata* KUE100 wild-type interaction with each epithelial cell was studied in terms of not only maximum adhesion force, work of adhesion, and rupture distance but also regarding the number of jumps and tethers, which were measured on the force-distance curves and plotted as repartition histograms. Mean value and their SD were extracted from a gaussian fit. A jump translates the attachment of the yeast cell probe to an adhesive unit, generally initially coupled to the cytoskeleton, following its unfolding, and finally its rupture[32]. In turn, a tether consists basically on the formation of a membrane nanotube, which is extruded and elongated, resulting in a long plateau due to the pulling of the membrane reservoir[32–36]. Membrane nanotubes have been found in epithelial cells[37] and other cell types[33], and are considered important for cell-to-cell adhesion and inter-cellular communication[38, 39]. The formation of tethers is affected by the actin cytoskeleton and glycocalyx connected to the membrane[33]. With just 5 s of contact time between cells, interaction forces can be measured, showing a maximum adhesion force of about 0.55 nN, work of adhesion of $5.98 \times 10^{-15}$ J, rupture distance of ~35 μm and an average of 1.63 jumps per force curve and 2.55 tethers per force–distance curve (Fig. 3a–e, respectively). The force–distance curves of the interaction reveal the presence of jumps and tethers, and generally long distances for the rupture of the interaction (Fig. 3f). This analysis results from the measurement of at least 4 yeast cells, from at least 3 independent cultures, recording around 64 force–distance curves per yeast cell.

The overall results show that, in general, different contact times between the yeast and epithelial cells do not alter adhesion force, work of adhesion, rupture distance, or number of tethers (Fig. 3a–c, e, f). These results are very surprising given that the presence of jumps suggests a role developed by adhesins in the interaction between yeast and epithelial cell, and therefore, a specific biologic interaction, expected to depend on contact time. It is possible that the adhesin/s at play have a remarkably small kinetic association constant ($K_{on}$) and the differences rely between 0 and 1 s of contact time. Interestingly, the main aspect that seems to change with contact time for *C. glabrata* is the number of jumps (Fig. 3d). Jumps have previously been described to translate the involvement of integrins in the adhesion process, especially when the number of jumps is higher than the number of tethers[36]. In fact, upon 30 s of contact time *C. glabrata* presents slightly higher numbers of jumps per force curve, which might indicate a role of adhesins in these first moments of the

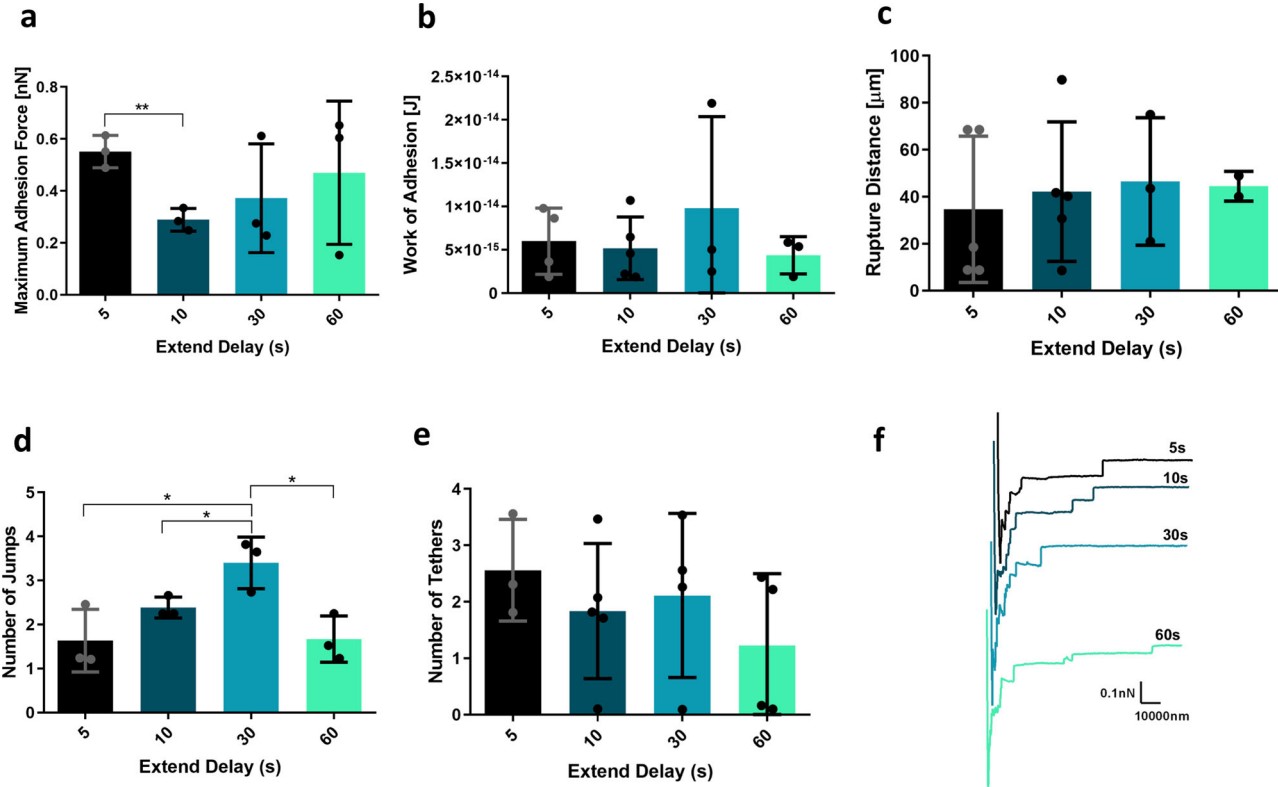

**Fig. 3 Interaction of *C. glabrata* wild-type strain KUE100 with human vaginal epithelial VK2/E6E7 cells by SCFS, using different contact times: 5, 10, 30, and 60 s.** Characterization of these interactions is based on the: **a** maximal adhesion force, **b** work of adhesion, **c** rupture distance, **d** number of jumps, and **e** number of tethers measured on each retraction curve. **f** Representative force–distance curves are presented for each contact time (the lighter the force curve, the higher the contact time). Horizontal lines indicate the average levels from at least 4 yeast cells, from at least 3 independent cell cultures, immobilized on the cantilever for the interaction with epithelial cells and 64 or 16 force–distance curves were recorded, for 5 and 10 s, and 30 and 60 s, respectively. Error bars indicate SDs. *$P < 0.05$; **$P < 0.01$, $n \geq 3$, except for the 60 s value in **c** ($n = 2$).

interaction with epithelial cells. Nevertheless, for other contact times, the differences are not clear between the number of jumps and tethers (Fig. 3d, e).

Comparing the interaction of *C. glabrata* and human vaginal epithelial cells with the interaction of *C. glabrata* and the plastic materials, the adhesion force measured on the interaction with plastic surfaces was found to be up to 40-fold higher (Supplementary Fig. 2a). Interestingly, the work of adhesion does not change significantly between the two types of interaction, except regarding the interaction of *C. glabrata* with polyvinyl chloride (Supplementary Fig. 2b). Rupture distance analysis shows the exact opposite results, the interaction of wild-type *C. glabrata* shows much higher average values of rupture distance than the interaction of *C. glabrata* with any of the materials tested (Supplementary Fig. 2c). The differences between *C. glabrata* adhesion to the materials and epithelial cells is also clear by the comparison between the representative force–distance curves of each case (Supplementary Fig. 2d, e).

**CgEfg1 and CgTec1 are involved in *C. glabrata* biofilm formation.** Although in *Candida albicans* a lot is already known about the major regulators of biofilm formation, very little has been uncovered for *C. glabrata*. According to the work of Nobile et al.[40], the regulatory network of biofilm formation in *C. albicans* is composed by CaEfg1, CaTec1, CaRob1, CaNdt80, CaBcr1, and CaBrg1. To unravel possible transcription factors for biofilm formation in *C. glabrata*, we studied the predicted orthologs in *C. glabrata*, of these transcription factors: CgEfg1 (CAGL0M07634g), CgEfg2 (CAGL0L01771g),

CgTec1 (CAGL0M01716g), CgTec2 (CAGL0F04081), CgNdt80 (CAGL0L13090g), and Bcr1 (CAGL0L00583g). No orthologs were found in the *C. glabrata* genome for CaRob1 or CaBrg1. Deletion mutants for the selected genes were built and tested for the ability to form 24 h biofilms on a polystyrene surface, with Sabouraud's dextrose broth (SDB) medium, pH 5.6, using the Presto Blue Cell Viability Assay. From this screening, only two genes, *CgEFG1* and *CgTEC1*, were found to be necessary for biofilm formation on these conditions (Fig. 4). The loss of either *CgEFG1* or *CgTEC1* decreases significantly the amount of biofilm formed, whereas their expression from a plasmid in the mutant background recovers the wild-type phenotype. The overexpression of either *CgEFG1* or *CgTEC1*, in addition, further increases the ability of wild-type cells to form biofilms (Fig. 4). The difference registered in terms of biofilm formation ability appears to be irrespective of the contribution of *CgEFG1* or *CgTEC1* to growth rate, as no growth defect was found to be displayed by the deletion mutants, when compared to the wild-type parental strain (Supplementary Fig. 3). These results indicate a clear role of CgTec1 and CgEfg1 in biofilm formation in *C. glabrata*, but also suggest that other regulators must yet be found and are probably different from the ones responsible for *C. albicans* biofilm formation.

In addition, the effect of the deletion of *CgEFG1* or *CgTEC1* in the ability to control ECM composition during conditions that promote biofilm formation was evaluated, particularly in what concerns protein and polysaccharide content. Upon 48 h of biofilm formation, the total protein and polysaccharide content of the ECM was evaluated in wild-type and in Δ*cgefg1* and Δ*cgtec1* mutant cells (Supplementary Fig. 4). The deletion of *CgEFG1* was

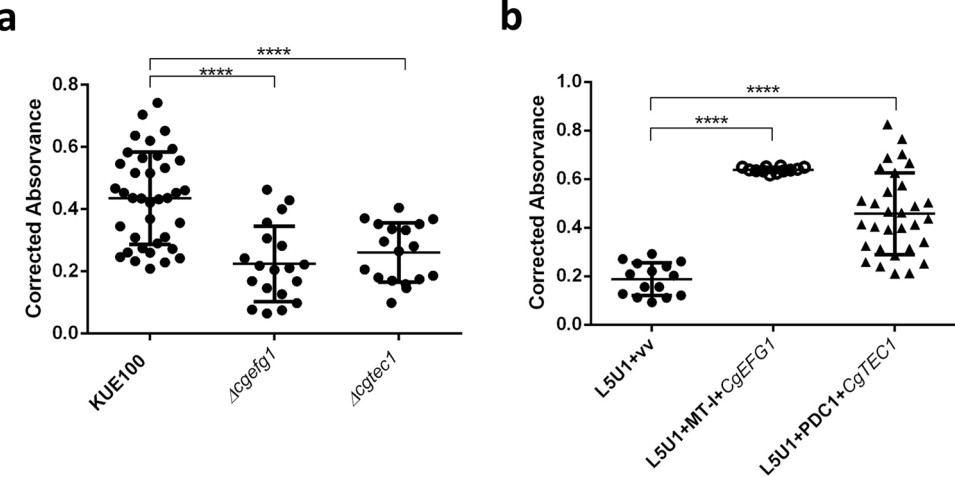

**Fig. 4 CgEfg1 and CgTec1 are necessary for *C. glabrata* biofilm formation on polystyrene surface.** Assessment of 24 h biofilm formation was performed by Presto Blue Cell Viability Assay in microtiter plates of: **a** *C. glabrata* wild-type KUE100 and deletion mutant Δ*cgefg1* strains harboring the pGREG576 cloning vector (vv), or the pGREG576_MT-I_*CgEFG1* (*CgEFG1*) plasmid, and **b** *C. glabrata* wild-type KUE100 and deletion mutant Δ*cgtec1* strains harboring the pGREG576 cloning vector (vv) or the pGREG576_PDC1_*CgTEC1* (*CgTEC1*) plasmid grown in SDB medium, pH 5.6. Values obtained are the average levels from at least three independent experiments. Error bars indicate SDs. ****$P < 0.0001$, $n \geq 16$.

found to lead to a 50% decrease in the polysaccharide content, while having no effect in the total protein content of the ECM. On the other hand, *CgTEC1* deletion was found not to affect the polysaccharide content of the biofilm ECM, but to lead to a 50% decrease in the total protein content. Altogether, these results suggest that each transcription factor plays a unique role in the control of biofilm formation in *C. glabrata*.

**CgEfg1 and CgTec1 are involved in *C. glabrata* adherence to the human vaginal epithelium.** Given the importance of CgEfg1 and CgTec1 in biofilm formation in *C. glabrata*, their involvement in *C. glabrata*'s adherence capacity to the human vaginal epithelial VK2/E6E7 cell line was also investigated. Adhesion of wild-type *C. glabrata* cells and of Δ*cgefg1* and Δ*cgtec1* mutant cells was evaluated after 30 min of contact, at 37 °C, 5% $CO_2$, with a multiplicity of infection (MOI) of 10. The results are presented as a percentage of adhered cells (ratio between the colony-forming unit (CFU)/ml after incubation with the epithelial cells and the initial CFU/ml estimated for each suspension). The deletion of *CgEFG1* or *CgTEC1* in *C. glabrata* is accompanied by a decrease in its capacity to adhere to the human vaginal epithelium (Fig. 5a). On the other hand, the overexpression of *CgEFG1* or *CgTEC1* genes in the *C. glabrata* wild-type strain L5U1 led to an increase in the adherence capacity of *C. glabrata* (Fig. 5b). These results provide evidence that both CgTec1 and CgEfg1 have important roles in *C. glabrata* adhesion to the human vaginal epithelium, evidencing a new role for CgTec1, which is not found for CaTec1, but a similar one between the Efg1 transcription factors of both species[41].

**Transcriptome-wide changes of *C. glabrata* cells upon biofilm formation.** Going further on the analysis of the role of CgEfg1 and CgTec1 transcription factors in the control of biofilm formation in *C. glabrata*, an RNA-seq analysis was performed to measure the transcriptome-wide remodeling occurring after 24 h of biofilm formation, in comparison to planktonic cultivation, in the wild-type and the two deletion mutant cells: Δ*cgefg1* and Δ*cgtec1*. This analysis enabled the identification of the transcriptional remodeling that underlies adaptation to biofilm versus planktonic growth by *C. glabrata* cells, as well as to identify the

genes whose expression is affected by the CgEfg1 and/or CgTec1 transcription factors.

Incredibly, upon 24 h of *C. glabrata* KUE100 biofilm formation, a total of 3072 genes exhibit differential expression, when compared to planktonic growing cells (Supplementary Data 2), which represents approximately half of the whole transcriptome. Within this total number, 1567 genes were found to be upregulated and 1505 genes downregulated. In order to see where the main differences lay, we organized the upregulated genes according to the biological processes they are related to (Supplementary Fig. 5). Biofilm formation requires the upregulation of several functional groups, including RNA metabolism and translation, carbon and energy metabolism, cell cycle, response to stress, amino acid metabolism, lipid metabolism, cell wall organization, adhesion, and biofilm formation.

Adhesion to surfaces and between cells is the first step of biofilm formation, being required to maintain the mature biofilm[1, 42]. Thus, the observed upregulation of genes related to adhesion was clearly expected. These include the following: *CgEPA1*, *CgEPA2*, *CgEPA3*, *CgEPA9*, *CgEPA10*, *CgEPA12*, *CgEPA20*, and *CgEPA23* encoding cluster I adhesins; *CgPWP1*, *CgPWP2*, *CgPWP3*, and *CgPWP5* encoding cluster II adhesins; *CgAED1* and *CgAED2* encoding cluster III adhesins; and *CgAWP1*, *CgAWP3*, *CgAWP4*, *CgAWP6*, and *CgAWP13* encoding cluster IV adhesins. Interestingly, *CgTEC1* and *CgEFG1* were also found to be upregulated in biofilm cells, but not their predicted paralogs *CgEFG2* or *CgTEC2*. This observation is consistent with the role played by the *C. glabrata* CgEfg1 and CgTec1, but not by CgEfg2 and CgTec2, in biofilm formation, as described in this study.

Interestingly, also among the upregulated genes in biofilm cells are a number of genes encoding multidrug resistance (MDR) transporters from the Major Facilitator Superfamily (MFS), previously shown to confer antifungal resistance in *C. glabrata*, including *CgQDR2*[43], *CgAQR1*[44], *CgTPO1_2*[45], and *CgTPO3*[46], along with the uncharacterized Open Reading Frames (ORFs) *CAGL0B02343g* and *CAGL0J00363g*, which are predicted MFS-MDR transporters.

**CgEfg1 and CgTec1 transcriptional control upon biofilm formation.** The absence of *CgEFG1* gene in *C. glabrata*, upon 24 h of biofilm formation on a polystyrene surface leads to 1164 genes

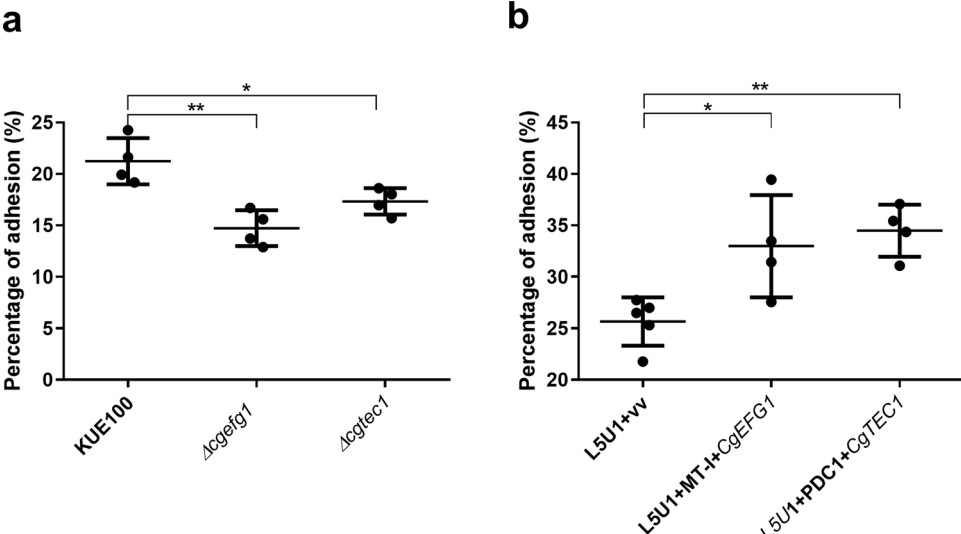

**Fig. 5 CgEfg1 and CgTec1 are necessary for *C. glabrata* adhesion to the VK2/E6E7 human vaginal epithelium cell line.** Adhesion of: **a** the *C. glabrata* parental KUE100, and Δ*cgefg1* and Δ*cgtec1* strains, and **b** the *C. glabrata* L5U1 strain harboring the pGREG576 cloning vector (vv) and the same strain harboring the pGREG576_MT-I_*CgEFG1* (*CgEFG1*)or pGREG576_PDC1_*CgTEC1* (*CgTEC1*) plasmids, to the human vaginal epithelial cells for 30 min at 37 °C under 5% $CO_2$. Values are averages of results from at least three independent experiments. Error bars represent SDs. *$P < 0.05$; **$P < 0.01$, $n \geq 4$.

being differentially expressed. CgEfg1 was found to activate 650 genes and repressed 514 genes, directly or indirectly (Supplementary Data 3). In turn, in 24 h *C. glabrata* biofilm cells, CgTec1 transcription factor was found to control a total of 1082 genes, activating 487 genes and repressing 595 genes, directly or indirectly (Supplementary Data 4). Out of the 1567 genes found to be upregulated in biofilm cells, 466 are regulated by CgEfg1 and 389 are regulated by CgTec1. The significant number of genes activated and repressed by both transcription factors highlights their impact as regulators of biofilm formation in *C. glabrata*.

When grouping the genes activated by CgEfg1 by function (Supplementary Fig. 6), it is possible to observe that the main functional groups being activated by this transcription factor upon biofilm formation are the following: unknown function, protein metabolism, response to stress, amino acid metabolism, carbon and energy metabolism, lipid metabolism and adhesion, and biofilm formation. When performing the same classification for the genes activated by CgTec1 (Supplementary Fig. 7), we observe that the main functional groups are the following: unknown function, response to stress, protein metabolism, cell cycle, carbon and energy metabolism, and cell wall organization.

Although activating shared functional groups, some seem to be differentially regulated by each transcription factor, suggesting that each one has their own target biological processes. For instance, CgEfg1 seems to contribute to the regulation of important genes for adhesion and biofilm formation, activating 13 of the 19 adhesin-encoding genes activated upon 24 h of biofilm formation: *CgPWP5*, *CgAED1*, *CgAED2*, *CgAWP13*, *CgAWP3*, *CgAWP1*, *CgPWP1*, *CgAWP4*, *CgPWP3*, *CgEPA9*, *CgEPA10*, *CgEPA12*, and *CgEPA20* (Supplementary Table 1). Interestingly, CgEfg1 also activates the expression of the genes encoding CgTec1 and CgBcr1 transcription factors, a behavior similar to what has been described for the biofilm network set in place in *C. albicans*[40]. In turn, CgTec1 seems to be more relevant to other important functional groups upon biofilm formation, such as the cell wall organization, cell cycle and invasive/filamentous growth, and virulence.

**CgEfg1 and CgTec1-activated adhesins are required for biofilm formation**. In order to confirm the previous observations and

discern at which stage of biofilm formation CgEfg1 and CgTec1 play a more significant role, the expression of *CgPWP5*, *CgAED2*, and *CgAWP13* adhesin-encoding genes was assessed at planktonic conditions and at 6, 24, and 48 h of biofilm formation, in the KUE100 wild-type strain and deletion mutants Δ*cgefg1* and Δ*cgtec1* cells (Fig. 6a–c). These adhesin-encoding genes were selected given their upregulation upon biofilm formation and activation by both transcription factors (Supplementary Data 2). Also, two of these adhesins, CgPwp5 and CgAed2, were found to be necessary for 24 h biofilm formation on a polystyrene surface (Fig. 6d). CgAwp13 had already been identified as an abundant protein present on the cell wall of a very polystyrene-adherent clinical isolate[47], but herein, the individual deletion of *CgAWP13* was found not to lead to a decrease in *C. glabrata* biofilm formation.

The three tested adhesins are particularly upregulated in later stages of biofilm formation (24 and 48 h), suggesting that they may be more relevant in yeast cell to yeast cell adhesion than to the initial steps of adhesion to polystyrene. Upon the deletion of CgEfg1 or CgTec1, the expression of *CgPWP5*, *CgAED2*, and *CgAWP13* genes is not downregulated in planktonic conditions nor upon 6 h of biofilm formation. Significantly, upon 24 and 48 h, the three adhesin-encoding genes are found to be strongly upregulated in the wild-type strain, but not in the Δ*cgefg1* or Δ*cgtec1* deletion mutants, indicating that both CgEfg1 and CgTec1 are relevant in later stages of biofilm formation (Fig. 6).

**CgEfg1, but not CgTec1, is involved in *C. glabrata* adhesion to plastic surfaces**. Given the results from the transcriptome-wide changes upon the deletion of *CgEFG1* or *CgTEC1* genes, indicating their importance in the control of different adhesin-encoding genes (Fig. 6), we explored their impact in the adhesion of *C. glabrata* to glass, polystyrene, silicone elastomer, and polyvinyl chloride. Just like that for the wild-type *C. glabrata* KUE100 strain, SCFS was applied with 5 s of contact between the yeast cell and surface, being the force–distance curves analyzed regarding maximum adhesion force, work of adhesion, and rupture distance (Figs. 7 and 8). The deletion of *CgEFG1* gene significantly affects the capacity of *C. glabrata* to adhere to any of the plastic surfaces tested, but no difference was found regarding

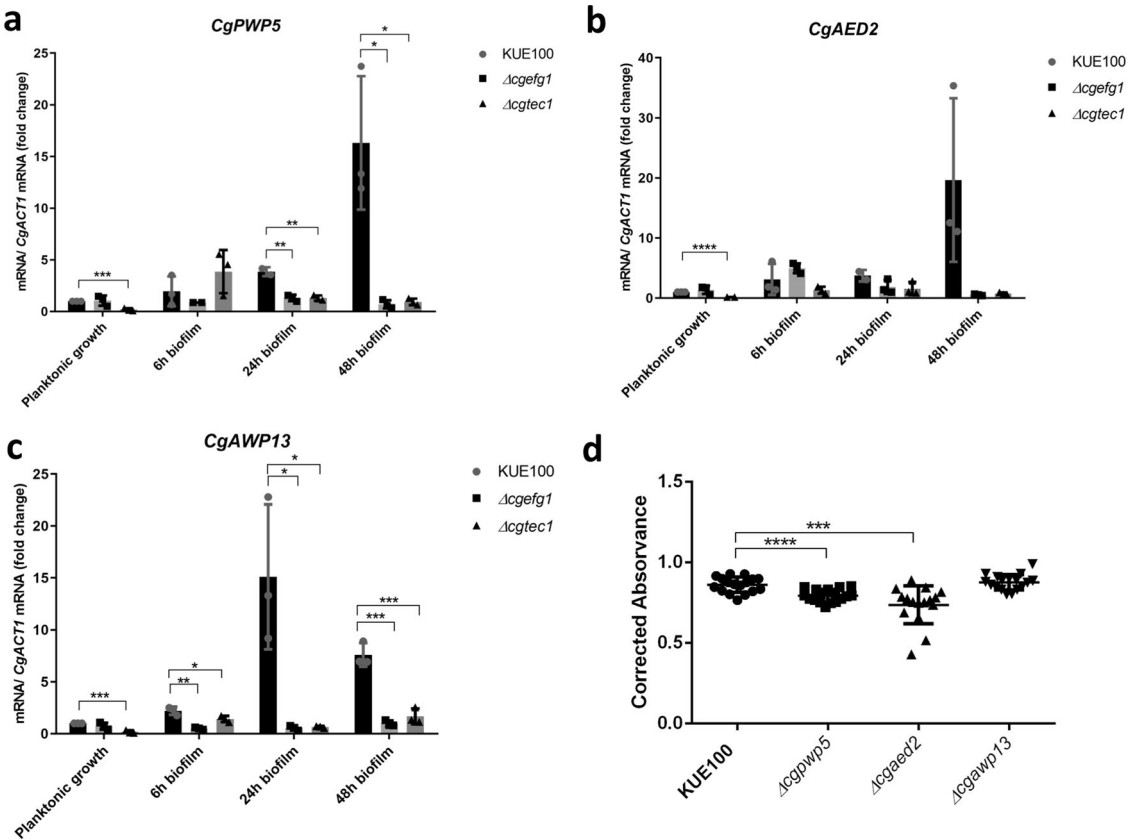

**Fig. 6 CgPwp5, and CgAed2 adhesins are necessary for biofilm formation, being regulated by CgEfg1 and CgTec1 transcription factors.** Shown are the transcript levels of: **a** CgPWP5, **b** CgAED2, and **c** CgAWP13 in the C. glabrata wild-type strain KUE100 and in the derived deletion mutants Δcgefg1 and Δcgtec1, in planktonic conditions and 6, 24, and 48 h of biofilm formation conditions on polystyrene surface in liquid SDB medium, pH 5.6. Transcript levels were assessed by quantitative RT-PCR, as described in "Methods." Values are averages of results from at least three independent experiments. Error bars represent SDs. **d** Twenty-four-hour biofilm formation quantified by Presto Blue Cell Viability Assay in microtiter plates of C. glabrata KUE100 wild-type and Δcgpwp5, Δcgaed2, and Δcgawp13 deletion mutant strains grown in SDB medium, pH 5.6. The data are displayed in a scatter dot plot, where each dot represents the level of biofilm formed in a sample. Horizontal lines indicate the average levels from at least three independent experiments. Error bars indicate SDs. *P < 0.05; **P < 0.01, ***P < 0.001; ****P < 0.0001, n ≥ 3.

the interaction with glass. C. glabrata maximum adhesion force towards polystyrene and polyvinyl chloride, and work of adhesion towards silicone elastomer and polyvinyl chloride, were found to decrease upon CgEFG1 deletion (Fig. 7a, b). Complementation of the wild-type KUE100 and deletion mutant Δcgefg1 with the pGREG576_MT-I_CgEFG1 increase the levels of maximum adhesion force to polystyrene and polyvinyl chloride (Fig. 7e, h), the deletion mutant recovering wild-type levels. These results clearly indicate that the changes of phenotype are directly due to the loss of CgEFG1 gene. In turn, no clear differences seem to exist between the KUE100 wild-type and the deletion mutant Δcgtec1 in terms of maximum adhesion force, work of adhesion, or rupture distance (Fig. 8).

**CgEfg1, but not CgTec1, is involved in C. glabrata adhesion to human vaginal epithelial cells**. As shown previously (Fig. 5), both CgEfg1 and CgTec1 are necessary to reach wild-type adherence levels upon 30 min of contact with human vaginal epithelium. To check whether they are also required for the initial steps of the adhesion process, which take place in seconds, the interaction forces established by Δcgefg1 and Δcgtec1 deletion mutant cells with a single epithelial cell were measured through SCFS, for a period of contact of 5 s, and compared to that of the parental wild-type strain. Upon the deletion of CgEFG1 gene, the maximum adhesion force, number of jumps and number of tethers of the interaction decrease significantly, which can be

depicted in the representative force–distance curves of the interaction of the wild-type and Δcgefg1 deletion mutant with an epithelial cell (Fig. 9). These results point out the importance of CgEfg1 in the capacity of C. glabrata to adhere to these epithelial cells, right at the initial point of interaction of the two cells. Complementation of the Δcgefg1 deletion mutant with the pGREG576_MT-I_CgEFG1 further increases the levels of maximum adhesion force and work of adhesion to epithelial cells (Fig. 7e, h), the deletion mutant recovering wild-type levels. These results clearly indicate that the changes of phenotype are directly due to the loss of CgEFG1 gene.

On the other hand, the loss of CgTEC1 gene did not result in such variance from the adherence profile of the wild-type strain (Fig. 10). It seems that CgTec1 transcription factor has a role in the adherence to the human vaginal epithelium but not as important as the role of CgEfg1, and does not contribute to the first moments of interactions between C. glabrata and a human vaginal epithelial cell.

## Discussion
C. glabrata is an effective pathogenic yeast, which uses adhesion and biofilm formation to better adapt to the environment and infect the human host. This work increases the understanding of the molecular basis of this process, from the transcriptome remodeling perspective to the nanoscale interaction level.

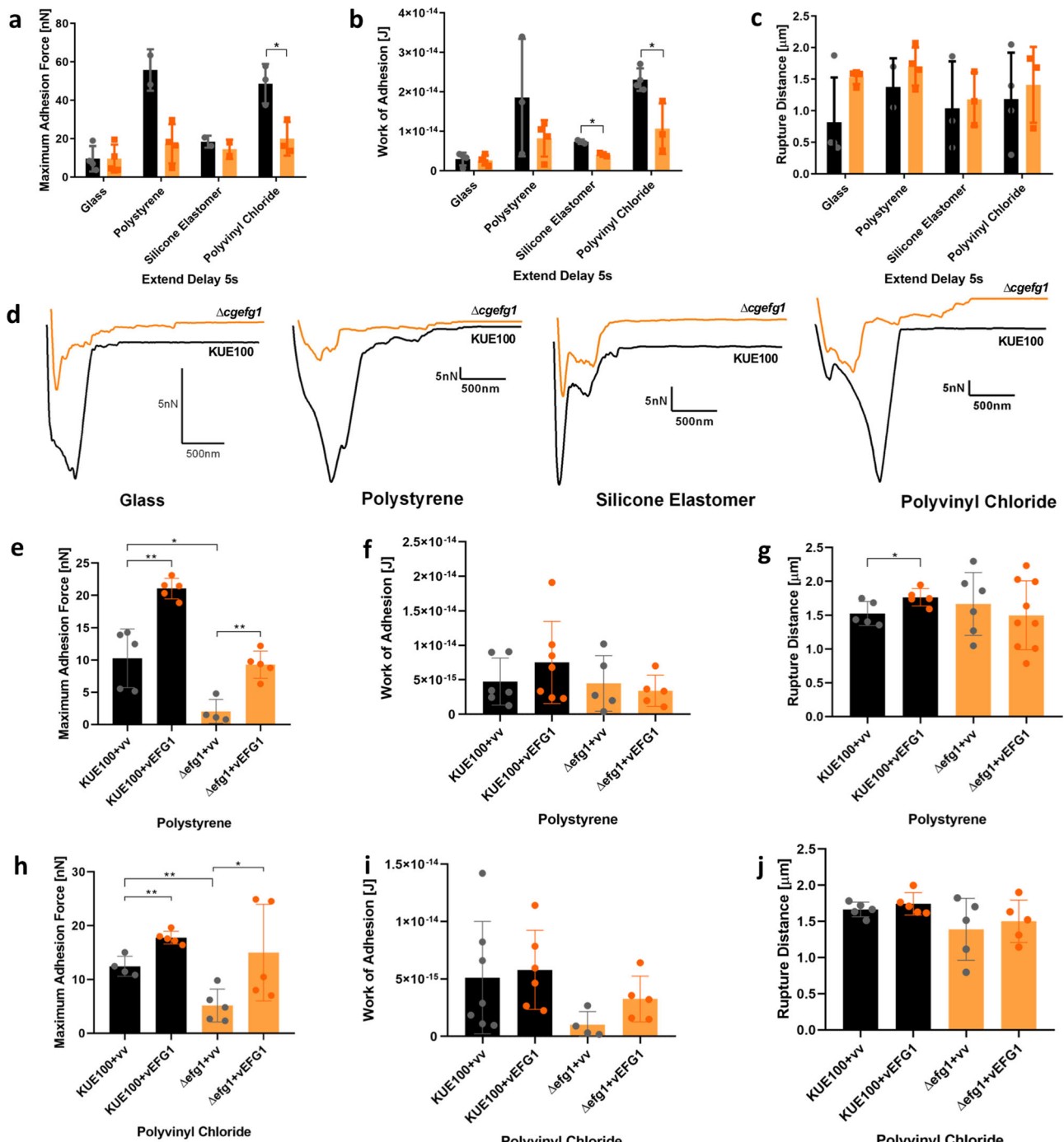

**Fig. 7 CgEfg1 is necessary for the adhesion of *C. glabrata* to polystyrene, silicone elastomer, and polyvinyl chloride, with 5 s of contact time.** The values obtained are the average levels of *C. glabrata* wild-type KUE100 strain (black) and deletion mutant Δ*cgefg1* (orange) regarding **a** maximum adhesion force, **b** work of adhesion, and **c** rupture distance measured on each retraction curve. **d** Representative force–distance curves of the interaction with glass, polystyrene, silicone elastomer, and polyvinyl chloride, by *C. glabrata* wild-type KUE100 (black) and Δ*cgefg1* deletion mutant single cells (orange). In addition, the average levels obtained for *C. glabrata* KUE100 and Δ*cgefg1* strains harboring the pGREG576 cloning vector (vv), or the pGREG576_MT-I_*CgEFG1 (CgEFG1)* plasmid regarding **e**, **h** maximum adhesion force, **f**, **i** work of adhesion, and **g**, **j** rupture distance measured on each retraction curve are also displayed. For every condition, at least four yeast cells, from at least three independent cell cultures, were immobilized on the cantilever for the interaction with each material. Twenty-five force–distance curves were recorded per yeast cell. Error bars indicate SDs. *$P < 0.05$; **$P < 0.01$, $n \geq 3$, except for the wt level in **a** ($n = 2$).

SCFS has allowed the in-depth study of the adhesion process of *C. glabrata* to several clinically relevant surfaces. The simplest adhesion of a single *C. glabrata* cell to glass was found to be higher than what has been found for other bacterial species such as *Staphylococcus epidermidis*, *Staphylococcus aureus*, and *Streptococcus mutans*, all ranging values of 4–5 nN of maximum adhesion force to borosilicate glass[48]. *C. glabrata*'s maximum adhesion force towards polystyrene, silicone elastomer, and polyvinyl chloride obtained (~14, ~10, and ~19 nN, respectively) was even higher than that obtained with glass, and the values found by Valotteau et al.[49], for *C. glabrata*

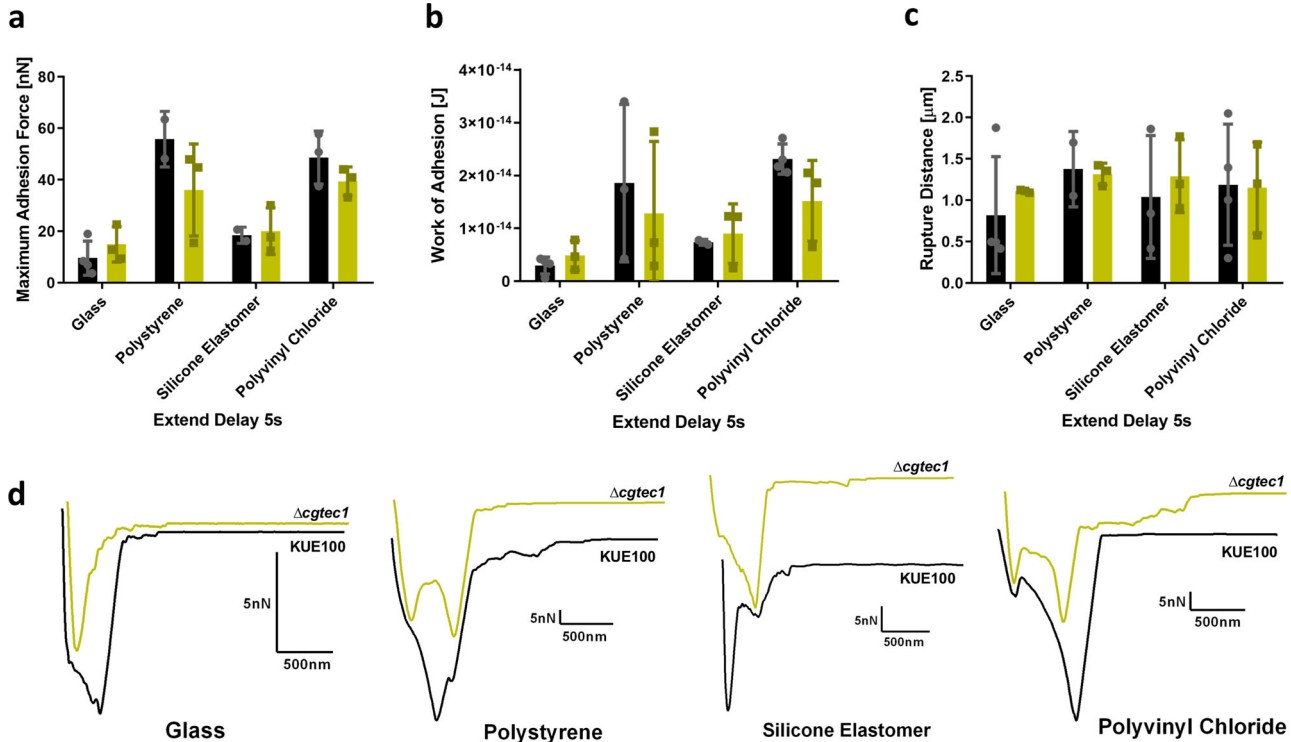

**Fig. 8 CgTec1 is not necessary for *C. glabrata* adhesion to polystyrene, silicone elastomer, and polyvinyl chloride, with 5 s of contact time.** Horizontal lines indicate the average levels of *C. glabrata* wild-type KUE100 strain (black) and deletion mutant *Δcgtec1* (green) regarding **a** maximum adhesion force, **b** work of adhesion, and **c** rupture distance measured on each retraction curve. **d** Representative force–distance curves of the interaction with glass, polystyrene, silicone elastomer, and polyvinyl chloride, by *C. glabrata* wild-type KUE100 (black) and *Δcgtec1* deletion mutant single cells (green). For every condition, at least four yeast cells, from at least three independent cell cultures, were immobilized on the cantilever for the interaction with each material. Twenty-five force–distance curves were recorded per yeast cell. Error bars indicate SDs, $n \geq 3$.

adhesion to hydrophobic methyl-terminated substrates (adhesion force of 0.3–0.5 nN and rupture distance of 0.2–0.4 μm). Nevertheless, it is lower than the interaction measured with other hydrophobic surfaces (~50 nN)[50]. *C. albicans* presents lower levels of maximum adhesion force (~40 nN) than *C. glabrata*, when comparing the same surfaces, whereas both present much higher adhesion forces than *Saccharomyces cerevisiae* (about 5 nN)[51], which is consistent with the role of adhesion in pathogenesis. However, when performing these comparisons, it is important to keep in mind that differences in adhesion have been reported between different *C. glabrata* strains. Strain to strain variation can be related to the presence in the genome of different adhesins, but also to differences in adhesin expression, which, e.g., can be related to strain differences in the function of the Sir Complex function or the Pdr1 transcription factor[52–54].

The interaction of *C. glabrata* with human vaginal epithelial cells (Fig. 3) revealed an adhesion profile similar to what has been found for other pathogens adhering to human epithelia. Similar to what we found, the interaction of *C. albicans* with macrophages exhibited a maximum adhesion force of 0.5 nN and the work of adhesion between 1 and $10 \times 10^{-15}$ J. Moreover, the study of force–distance curves also showed the presence of jumps and tethers[55]. In turn, the interaction of a single cell of human bronchial epithelium to a monolayer of *Pseudomonas aeruginosa* revealed an adhesion force of 0.67 nN and work of adhesion of about $4.14 \times 10^{-15}$ J[56], a bit higher than what is observed for *C. glabrata* or *C. albicans*. Although considering different microorganisms and epithelial cells, the values obtained are not that far apart from one another, clearly indicating the capacity of these pathogens to adhere to human cells, an important characteristic in their colonization and infection ability.

The dramatically increased capacity of *C. glabrata* to adhere to medical materials when compared to epithelial cells (Fig. 5) correlates with the difficulty to eradicate *C. glabrata* from the human host, when it is attached to medical devices compared to when it is just interacting with host cells. This highlights the need to change the materials used in medical devices, so that they are not a perfect match for adherence to *C. glabrata* and other pathogens.

To further pursuit the study of biofilm formation, we performed a genetic screening that identified CgEfg1 and CgTec1 transcription factors as necessary to biofilm formation and adhesion to the human vaginal epithelium. The *C. glabrata* Tec1 and Efg1 proteins described in this study are orthologs of well-characterized transcription factors, one each in *C. albicans* (Tec1 and Efg1) and one each in *S. cerevisiae* (Tec1 and Sok2), according to the *Candida* Genome Database (CGD), and based on the Yeast Genome Order Browser and the *Candida* Genome Order Browser databases, supporting the strong hypothesis that they act indeed as transcription factors as well in *C. glabrata*. This screening also revealed that the regulatory network of *C. glabrata* controlling biofilm formation is different from that described for *C. albicans*[40]. Nevertheless, upon biofilm formation, both species suffer major alterations in the transcriptome, *C. glabrata* with 3072 genes with altered expression and *C. albicans* with 2235 genes differentially expressed, comparatively to planktonic conditions, according to Nobile et al.[40]. In both species, genes differently expressed are related to adhesion and metabolism[57]. Adhesion allows the starting point and sustenance of the biofilm, being important for biofilm formation. Regarding metabolism, our data set shows upregulation of biosynthesis of positively

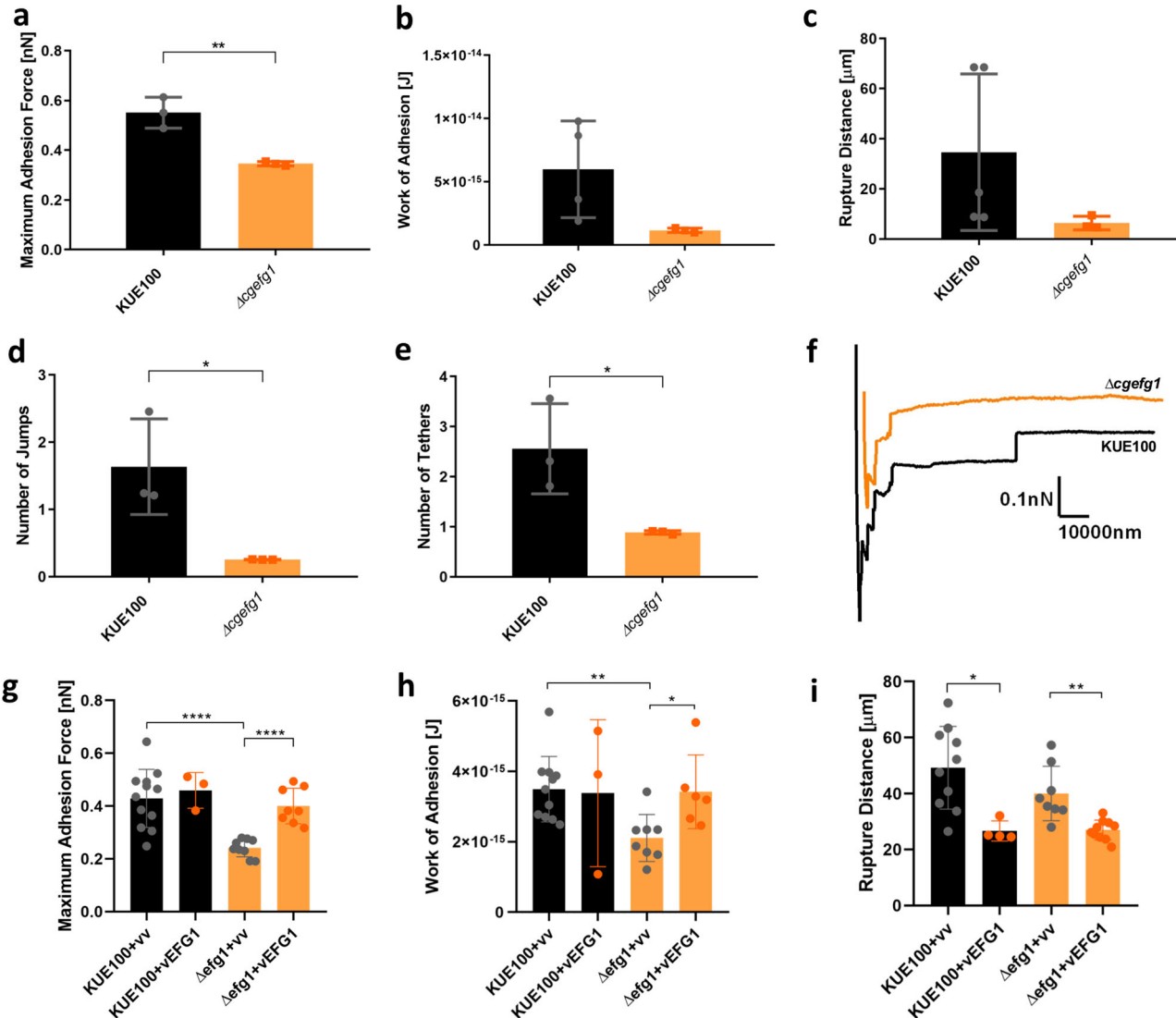

**Fig. 9 Interaction of *C. glabrata* wild-type strain KUE100 (black) and deletion mutant Δ*cgefg1* (grey) with human vaginal epithelial VK2/E6E7 cells by SCFS, using 5s of contact time.** Characterization of these interactions is based on the **a** maximal adhesion force, **b** work of adhesion and **c** rupture distance, **d** number of jumps and **e** number of tethers measured on each retraction curve. **f** Representative force-distance curves of the interaction of *C. glabrata* wild-type KUE100 (black) and Δ*cgefg1* deletion mutant single cells (orange) with epithelial cells. Additionally, the average levels obtained for *C. glabrata* KUE100 and Δ*cgefg1* strains harbouring the pGREG576 cloning vector (vv), or the pGREG576_MT-I_*CgEFG1* (*CgEFG1*) plasmid regarding **g**, maximum adhesion force, **h**, work of adhesion, and **i**, rupture distance measured on each retraction curve, are also displayed. Horizontal lines indicate the average levels from at least 4 yeast cells, from at least 3 independent cultures, immobilized on the cantilever for the interaction with epithelial cells 64 force distance curves were recorded. *, P<0.05; **, P<0.01, ****, P<0.0001. Error bars indicate SDs, $n \geq 3$.

charged amino acids, indicating that the cells are suffering from nitrogen limitation. Moreover, 13 out of the 18 genes required for the biogenesis of peroxisomes, where fatty acid β-oxidation occurs, were found to be upregulated upon *C. glabrata* biofilm formation, suggesting biofilm cells might also be experiencing carbon source limitation. An indication also found by Fox et al.[57] study on transcriptomics modulation upon biofilm formation.

The importance of CgEfg1 and CgTec1 in the control of biofilm formation is very clear given the number of genes differently expressed upon biofilm formation and controlled by each transcription factor, including adhesin-encoding genes *CgAWP13*, *CgPWP5*, and *CgAED2*, found to be activated especially in mature biofilms, the two latter being demonstrated to be required for biofilm formation. It seems from our data sets that both CgEfg1 and CgTec1 have an important role contributing to the production of ECM and biofilm proteins, given the activation of genes belonging to the functional groups of protein metabolism, and

carbon and energy metabolism. In addition, CgEfg1 and CgTec1 were found to control polysaccharide and protein composition of the ECM, respectively. Although CgEfg1 controls a lot of genes regarding adhesion, CgTec1 seems to be more relevant to cell wall organization, cell cycle and invasive/filamentous growth, and virulence. Such results suggest that CgTec1 might be involved in the control of pseudohyphae formation, similar to one of the roles of its *C. albicans* ortholog[58].

When evaluating the capacity of wild-type and of Δ*cgefg1* and Δ*cgtec1* deletion mutants to adhere to clinically relevant surfaces by SCFS, agreeing results were found. Although CgTec1 is not necessary for the first moments of adhesion to plastic surfaces or epithelial cells, CgEfg1 was found to have a significant role. The clear importance of CgEfg1 in *C. glabrata* adhesion is most likely related to its role in the activation of 13 adhesin-encoding genes, upon 24 h of biofilm formation. Such influence on adhesion is likely extended to the beginning of biofilm formation, given that

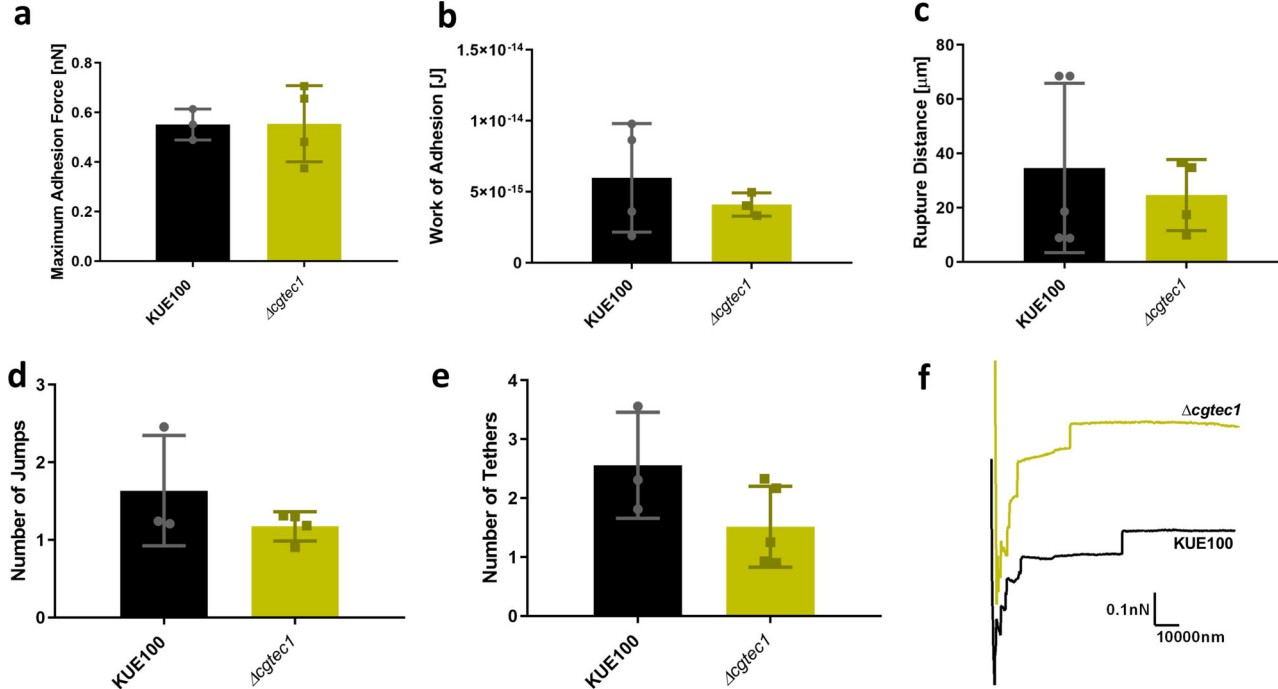

**Fig. 10 Interaction of *C. glabrata* wild-type strain KUE100 (black) and deletion mutant Δ*cgtec1* (green) with human vaginal epithelial VK2/E6E7 cells by SCFS, using 5 s of contact time.** Characterization of these interactions is based on the: **a** maximal adhesion force, **b** work of adhesion, **c** rupture distance, **d** number of jumps, and **e** number of tethers measured on each retraction curve. **f** Representative force–distance curves of the interaction of *C. glabrata* wild-type KUE100 (black) and Δ*cgtec1* deletion mutant single cells (green) with epithelial cells. Horizontal lines indicate the average levels from at least four yeast cells, from at least three independent cultures, immobilized on the cantilever for the interaction with epithelial cells. Sixty-four force–distance curves were recorded. Error bars indicate SDs, *n* ≥ 3.

upon *CgEFG1* deletion, but not *CgTEC1* deletion, a decrease in the expression of *CgAWP1*, *CgAWP4*, *CgEPA6*, *CgEPA7*, and *CgEPA8* adhesin-encoding genes was observed in planktonic cells (Supplementary Table 1), a feature that may underlie the differential ability of these cells to adhere to biotic and abiotic surfaces in their first contact. To corroborate this idea, the work of El-Kirat-Chatel et al.[50] show that Δ*cgepa6* deletion mutant has decreased adhesion force to hydrophobic surfaces when compared to the wild type, suggesting that CgEpa6 is one of the adhesins controlled by CgEfg1, responsible for its importance in *C. glabrata*'s first moments of adhesion to the plastic surfaces studied herein.

Overall, this work describes two major regulators of adhesion and biofilm formation, CgEfg1 and CgTec1, constituting promising targets for the development of new therapeutic strategies to fight *C. glabrata*-promoted candidiasis.

## Methods

**Strains and growth medium**. *C. glabrata* CBS138, KUE100[59], and L5U1 (cgura3Δ0 cgleu2Δ0) strains, the later kindly provided by John Bennett, NIAID, NIH, Bethesda, were used in this study. Cells were batch-cultured at 30 °C with orbital agitation (250 r.p.m.) in the following growth media. Yeast extract Peptone Dextrose (YPD) growth media, containing per liter: 20 g glucose (Merck), 10 g yeast extract (Difco), and 20 g bacterial-peptone (LioChem). Basal minimal (BM) minimal growth medium contained per liter: 20 g glucose (Merck), 2.7 g (NH₄)₂SO₄ (Merck), and 1.7 g yeast nitrogen base without amino acids or (NH₄)₂SO₄ (Difco). SDB contained 40 g glucose (Merck) and 10 g peptone (Lio-Chem) per liter.

The VK2/E6E7 human epithelium cell line (ATCC® CRL-2616™) was used for adhesion assays. This cell line is derived from the vaginal mucosa of healthy premenopausal female submit to vaginal repair surgery and immortalized with human papillomavirus 16/E6E7. Cell maintenance was achieved with KSF medium, containing 0.1 ng/mL human recombinant epidermal growth factor, 0.05 mg/mL bovine pituitary extract, and additional 44.1 mg/L calcium chloride. Cells were maintained at 37 °C, with 95% air and 5% CO₂.

**Cloning of the *C. glabrata* *CgEFG1* and *CgTEC1* genes (ORF *CAGL0M07634g* and *CAGL0M01716g*)**. The pGREG576 plasmid from the Drag & Drop collection[60] was used to clone and express the *C. glabrata* ORF *CAGL0M07634g* and *CAGL0M01716g* in *S. cerevisiae*, as described before for other heterologous genes[61]. pGREG576 was acquired from Euroscarf and contains a galactose inducible promoter (*GAL1*) and the yeast selectable marker *URA3*. The *CgEFG1* and *CgTEC1* genes were generated by PCR reaction, using genomic DNA extracted from the sequenced CBS138 *C. glabrata* strain, with the primers present in Supplementary Table 2. To enable expression of the *CgTEC1* gene in *C. glabrata*, the *GAL1* promoter was replaced by the constitutive *PDC1* *C. glabrata* promoter, whereas for *CgEFG1* the replacement was performed using the copper-induced MT-I *C. glabrata* promoter. The PDC1 and MT-I promoters DNA was generated by PCR, using the primers in Supplementary Table 2. The recombinant plasmids pGREG576_*CgEFG1*, pGREG576_*CgTEC1*, pGREG576_MT-I_*CgEFG1*, and pGREG576_PDC1_*CgTEC1* were obtained through homologous recombination in *S. cerevisiae* and verified by DNA sequencing.

**Disruption of the *C. glabrata* *CgEFG1*, *CgTEC1*, and *CgURA3* genes (ORF *CAGL0M07634g*, *CAGL0M01716g*, and *CAGL0I03080g*)**. The deletion of the *CgEFG1* and *CgTEC1* genes was carried out in the parental strain KUE100, using the method described by Ueno et al.[59]. The target genes were replaced by a DNA cassette including the *CgHIS3* gene, through homologous recombination. The pHIS906 plasmid including *CgHIS3* was used as a template and transformation was performed using the Lithium Acetate method, as described previously.[59] Briefly, cells were incubated in 10 ml of YPD liquid medium and cultured overnight with shaking at 37 °C. The cells were resuspended in 10 ml of fresh YPD up to an OD600 of 1.0, ~2–3 h at 37 °C, and collected by centrifugation. Upon rinsing with TE buffer (1 M Tris-HCl and 0.2 M EDTA, pH 8.0), cells were resuspended in 10 ml of 0.15 M lithium acetate dissolved in TE buffer (LiOAc/TE) and shaken lightly for 1 h at 30 °C. The cells were again collected and resuspended in 400 μl of 0.15 M LiOAc. Sixty microliters of the cell suspension was then supplemented with 5–10 μg of the disruption cassettes, 20 μg denatured salmon sperm DNA (Wako), polyethylene glycol 4000 and 0.15 M LiOAc, and incubated for 45 min at 37 °C. After mixing carefully, the cells were heat shocked by incubating for 45 min at 42 °C and then spread onto appropriate selection plates and incubated at 37 °C for a few days.[59] Recombination locus and gene deletion were verified by PCR using the primers indicated in Supplementary Table 2.

The disruption of the *C. glabrata* URA3 gene encoded by ORF *CAGL0I03080g* was carried out in the KUE100_Δ*cgefg1* mutant, using the CRISPR-Cas9 system from Vyas et al.[62] Briefly, a *CgURA3* guide RNA (gRNA) sequence selected from

the resources made available by Vyas et al.[62] was cloned in the pV1382 plasmid, previously linearized with the restriction enzyme BsmBI (NEB). The CgURA3 gRNA was obtained by oligonucleotide annealing and the product ligated into the previously linearized pV1382 plasmid to obtain the pV1382 CgURA3 vector. The construct was verified by DNA sequencing. The plasmid was transformed into the KUE100_Δcgefg1 strain and cells were then directly plated on 5-Fluoroorotic acid to select for URA- cells. Sequential passages in nonselective medium (YPD) were performed to avoid detrimental effects of further Cas9 expression and CgURA3 loss of function was further confirmed by the inability to grow in medium without uracil. The introduction of pGREG576-derived plasmids in the edited strains was able to rescue the growth impairment in the absence of uracil.

**Biofilm quantification**. C. glabrata strains were tested regarding their capacity to form biofilm on a polystyrene surface, recurring to the Presto Blue Cell Viability Assay. For that, the C. glabrata strains were grown in SDB (pH 5.6) medium and collected by centrifugation at mid-exponential phase. The cells were inoculated with an initial $OD_{600nm} = 0.05 \pm 0.005$ in 96-well polystyrene microtiter plates (Greiner) in SDB (pH 5.6) medium. Cells were cultivated at 30 °C during $24 \pm 0.5$ h with mild orbital shaking (70 r.p.m.). After the incubation time, each well was washed two times with 100 μL of sterile phosphate-buffered saline (PBS) pH 7.4 (PBS contained per liter: 8 g NaCl (Panreac), 0.2 g KCl (Panreac), 1.81 g $NaH_2PO_4$. $H_2O$ (Merck), and 0.24 g $KH_2PO_4$ (Panreac), to remove the cells unattached to the formed biofilm. Then, Presto Blue reagent was prepared in a 1 : 10 solution in the medium used for biofilm formation, adding 100 μL of the solution to each well. Plates were incubated at 37 °C for 30 min. Afterwards, absorbance reading, at the wavelength of 570 and 600 nm for reference, was determined in a microplate reader (SPECTROstar Nano, BMG Labtech).

**Human vaginal epithelial cell adherence assay**. For the adhesion assays, VK2/ E6E7 human epithelium cells were grown and inoculated in 24-well polystyrene plates (Greiner) with a density of $2.5 \times 10^5$ cell/mL a day prior to use. In addition, C. glabrata cells were inoculated with an initial $OD_{600nm} = 0.05 \pm 0.005$, cultivated at 30 °C, during $16 \pm 0.5$ h, with orbital shaking (250 r.p.m.) in YPD medium. In order to initiate the assay, the culture medium of mammalian cells was removed and substituted by new culture medium in each well and, subsequently, C. glabrata cells were added to each well, with a density of $12.5 \times 10^5$ CFU/well, corresponding to a MOI value of 10. Then, the plate was incubated at 37 °C, 5% $CO_2$, for 30 min. Afterwards, each well was wash three times with 500 μL of PBS pH 7.4, following the addition of 500 μL of Triton X-100 0.5% and incubation at room temperature for 15 min. The cell suspension in each well was then recovered, diluted, and spread onto agar plates to determine the CFU count, which represent the proportion of adherent cells to the human epithelium.

**RNA sample extraction and preparation**. Cells were grown in SDB (pH 5.6) medium. Planktonic cells were cultured at 30 °C with orbital agitation (250 r.p.m.), whereas biofilm cells were cultured at 30 °C, in square Petri dishes, with orbital agitation (30 r.p.m.). Total RNA was extracted from wild-type and single deletion mutant cells during planktonic exponential growth and upon 24 h of biofilm growth. Total RNA was isolated using an Ambion Ribopure-Yeast RNA kit, according to the manufacturer's instructions.

**RNA-seq of C. glabrata cells in planktonic and biofilm conditions**. Strand-specific RNA-seq library preparation and sequencing was carried out as a paid service by the NGS core from Oklahoma Medical Research Foundation, Oklahoma City, Oklahoma, USA. Prior to RNA-seq analysis, quality-control (QC) measures were implemented. Concentration of RNA was ascertained via fluorometric analysis on a Thermo Fisher Qubit fluorometer. Overall quality of RNA was verified using an Agilent Tapestation instrument. Following initial QC steps, sequencing libraries were generated using the Illumina Truseq Stranded Total RNA library prep kit with ribosomal depletion via RiboZero Gold according to the manufacturer's protocol. Briefly, ribosomal RNA was depleted via pulldown with bead-bound ribosomal-RNA complementary oligomers. The RNA molecules were then chemically fragmented and the first strand of (cDNA) was generated using random primers. Following RNase digestion, the second strand of cDNA was generated replacing dTTP in the reaction mix with dUTP. Double-stranded cDNA then underwent adenylation of 3′-ends following ligation of Illumina-specific adapter sequences. Subsequent PCR enrichment of ligated products further selected for those strands not incorporating dUTP, leading to strand-specific sequencing libraries. Final libraries for each sample were assayed on the Agilent Tapestation for appropriate size and quantity. These libraries were then pooled in equimolar amounts as ascertained via fluorometric analyses. Final pools were absolutely quantified using quantitative PCR (qPCR) on a Roche LightCycler 480 instrument with Kapa Biosystems Illumina Library Quantification reagents. Sequencing was performed on an Illumina HiSeq 3000, producing $2 \times 150$ bp paired-end reads, 2 GB clean data, yielding 52 M reads per sample. Paired-end reads were obtained from wild-type (C. glabrata KUE100) and correspondent deletion mutant strains Δcgefg1 and Δcgtec1 (CAGL0M07634g and CAGL0M01716g). Two replicates of each sample were obtained from three independent RNA isolations, subsequently pooled together. Sample reads were trimmed using Skewer[63] and aligned to the C.

glabrata CBS138 reference genome, obtained from the CGD,[64] using TopHat[65] HTSeq[66] was used to count mapped reads per gene. Differentially expressed genes were identified using DESeq2[67] with an adjusted p-value threshold of 0.01 and a log2 fold-change threshold of −1.0 and 1.0. Default parameters in DESeq2 were used. Significantly differentially expressed genes were clustered using hierarchical clustering in R.[68] C. albicans and S. cerevisiae homologs were obtained from the CGD and Saccharomyces Genome Database (SGD),[69] respectively.

**Accession number**. The data sets were deposited at the Gene Expression Omnibus, NCBI database, with the reference number GSE140427.

**Transcriptomic analysis**. The RNA-seq analysis provided three data sets as follows: wild-type in planktonic growth vs. wild-type biofilm growth; wild-type vs. Δcgefg1 upon biofilm growth; and wild-type vs. Δcgtec1 upon biofilm growth. The genes of each data set were submitted to several analyses using different databases and bioinformatic tools, so they could be grouped according to their biological functions. This was accomplished mainly by resorting to the description of the C. glabrata genes found on the CGD (http://www.candidagenome.org). The uncharacterized genes were clustered based on the description of ortholog genes in S. cerevisiae or in C. albicans, according to the SGD or in CGD, respectively. Go-Stats from GoToolBox web server[70] allowed the determination of the main Gene Ontology terms to which the genes were related. From this organization, several genes related to cell adhesion were chosen for the following gene expression analysis.

**Single gene expression analysis**. The levels of CgPWP5, CgAED2, and CgAWP13 transcripts in the KUE100 wild type, Δcgefg1, and Δcgtec1 deletion mutant cells upon 6, 24, and 48 h of biofilm formation on polystyrene surface and planktonic conditions were assessed by quantitative real-time PCR. Cells were grown in SDB (pH 5.6) medium. Planktonic cells were cultured at 30 °C with orbital agitation (250 r.p.m.), while biofilm cells were cultured at 30 °C, in square Petri dishes, with orbital agitation (30 r.p.m.). Total RNA was extracted planktonic and biofilm conditions (6, 24, and 48 h of biofilm formation). Synthesis of cDNA for real-time reverse-transcriptase PCR (RT-PCR) experiments, from total RNA samples, was performed using the Multiscribe™ reverse-transcriptase kit (Applied Biosystems), following the manufacturer's instructions, and using 10 ng of cDNA per reaction. The RT-PCR step was carried out using NZY qPCR Green Master Mix (2×) (NZYTECH). Primers for the amplification of the CgPWP5, CgAED2, CgAWP13, and CgACT1 cDNA were designed using Primer Express Software (Applied Biosystems) (Supplementary Table 2). The RT-PCR reactions were conducted in a thermal cycler block (7500 Real-Time PCR System-Applied Biosystems). The CgACT1 mRNA level was used as an internal control[71].

**Quantification of ECM components**. Protein and polysaccharide content was quantified as described previously by Panariello et al.[72] For that, C. glabrata KUE100, and Δcgefg1 and Δcgtec1 deletion mutant biofilms were grown in microtiter plates of 24 wells with SDB medium pH 5.6, for 48 h, at 30 °C, 70 r.p.m., with a washing step at 24 h. After 48 h, C. glabrata biofilms were scrapped under a sterile solution of 0.89% NaCl and submitted to 1 min vortex at maximum speed, for mechanical disruption of the ECM. Following a centrifugation at $5500 \times g$, 10 min, 4 °C, the ECM components present on the supernatant were stored at −20 °C. For protein quantification, Bovine serum albumin solution (P5369; Sigma-Aldrich, St. Louis, MO, USA) was prepared in saline buffer and the following concentrations were used as a standard curve: 0.2, 0.39, 0.78, 1.56, 3.125, 6.25, 12.5, 25, and 35 μg/mL. In 96-well plates, 40 μL of Bradford reagent (B6916; Sigma-Aldrich) was mixed with 160 μL of each standard and sample solution. The reaction was carried out for 15 min and the absorbance measured at 595 nm in a spectrophotometer. For water-soluble polysaccharide content quantification, 2.5 volumes of 95% ethanol were added to 1 mL per sample of the homogenized supernatant. Polysaccharides were precipitated for 18 h at −20 °C and centrifuged $9500 \times g$ for 20 min at 4 °C. Afterwards, the supernatants were discarded and samples were washed three times with ice-cold 75% ethanol and left to air dry. Each pellet was resuspended with 1 mL of water and total polysaccharides were quantified using the phenol-sulfuric acid method[73]. Glucose was used for the standard curve (10, 30, 50, 70, 90, and 110 mg/mL). For the method, 200 μL of 5% phenol were added to a glass tube with 200 μL of the sample or standard curve point. After careful mixing, 1 mL of sulfuric acid was added to each glass tube under agitation. After 20 min of reaction, samples were measured at 490 nm in a spectrophotometer.

**Single-cell force spectroscopy**. For the interaction of C. glabrata strains with plastic surfaces and human vaginal epithelial cells, SCFS was implemented. Yeast cell probes were prepared by adding stationary-phase C. glabrata cells to the petri dish with the plastic materials attached by glue or to the petri dish with prepared epithelial cells. Triangular shaped tipless cantilevers (NP-O10, Microlevers, Bruker Corporation) were coated with conA. For the coating, the cantilevers were immersed overnight in 100 μg/mL of conA solution and washed in acetate buffer pH 5.2 before use. Single yeast cells placed on the petri dish were attached onto the conA-coated cantilevers using Nanowizard III AFM (Bruker, JPK BIOAFM), approaching the cantilever onto a single cell for 30 s. Interaction of C. glabrata cells

with plastic surfaces was followed with force measurements at room temperature (25 °C), under acetate buffer pH 5.2. For the interaction of *C. glabrata* cells with human vaginal epithelial cells, Nanowizard III AFM equipped with CellHesion module (Bruker, JPK BIOAFM) was used and VK2/E6E7 cells ($7 \times 10^4$) were platted on a petri dish under the KSF culture medium, at 37 °C, with 95% air and 5% $CO_2$ to grow overnight. Single-cell measurements were conducted under this environment, thanks to the petri dish heater (Bruker, JPK BIOAFM). The nominal spring constant (Kc) of the cantilevers used for the interaction with the plastic surfaces was ~0.35 and 0.06 N/m for the cantilevers used to measure interactions between yeast and epithelial cells. The cantilevers were all calibrated. Their sensitivity ranged from ~14 to 30 nm/V and their spring constants, determined by the thermal noise method, were in agreement with the manufacturer: the Kc ranged from 0.03 to 0.12 for the nominal 0.06 N/m and from 0.175 to 0.7 for the nominal 0.35 N/m. Several force–distance curves were recorded for an area of 10 μm × 10 μm of the plastic surface and applied force of 1 nN, and with an area of 3 μm × 3 μm of the epithelial cell and applied force of 0.5 nN. Cell probes were approached and retracted to the plastic surfaces with a speed of 5 μm/s and a contact time of 0, 0.5, 1, and 5 s, and to the human vaginal epithelial cells with a speed of 20 μm/s and 5, 10, 30, and 60 s of contact time. At least five yeast cells from at least three independent cultures were immobilized on the cantilever for the interaction of each material, and at least four yeast cells from at least three independent cultures were immobilized on the cantilever for the interaction with epithelial cells. Adhesion force, work of adhesion, rupture distance, number of jumps, and tethers histograms were obtained by calculating the maximum adhesion peak, the area under the curve, the last rupture distance, and counting the number of jumps and tethers for each force curve. Two hundred and fifty-six, 100, and 25 force–distance curves were recorded for the interaction with the materials, with 0, 0.5–1, and 5 s, respectively, and 64 and 16 force–distance curves were recorded for the interaction with epithelial cells, with 5 and 10 s, and 30 and 60 s, respectively.

**Atomic force microscopy.** Topographic images of VK2/E6E7 epithelial cells were obtained using the Nanowizard III AFM (JPK Instruments) coupled with an axiovert microscope from Zeiss with QI™ mode. VK2/E6E7 cells ($7 \times 10^4$) were platted on a petri dish under the KSF culture medium, at 37 °C, with 95% air and 5% $CO_2$ to grow overnight, following imaging under this environment. MLCT cantilevers (Bruker probes) with a spring constant of 0.012 N/m were used. The cantilevers spring constants were determined by the thermal noise method. QI™ settings used are the following: Z-length: 3 μm; applied force: 4 nN; speed: 150 μm/s. JPK data processing (JPK Instrument, Berlin, Germany) software was used for image processing as described before[30].

**Statistics and reproducibility.** Statistical analysis was performed using Graphpad Prism Software version 8.0 (La Jolla, CA, USA) and analyzed with Student's *t*-test, considering in all cases $n \geq 3$. *p*-values equal or inferior to 0.05 were considered statistically significant.

**Reporting summary.** Further information on research design is available in the Nature Research Reporting Summary linked to this article.

## Data availability
RNA-sequencing data sets were deposited at the Gene Expression Omnibus, NCBI database, with the reference number GSE140427. All other data are available from the corresponding author on reasonable request. Raw data underlying all figures have been provided as Supplementary Data 1.

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

## Acknowledgements

This work was supported by "Fundação para a Ciência e a Tecnologia" (FCT) (Contracts PTDC/BBB-BIO/4004/2014 and PTDC/BII-BIO/28216/2017, and BIOTECnico PhD grants to M.C., P.P., and A.I.P., as well as AEM PhD grant to R.V.). Funding received by iBB-Institute for Bioengineering and Biosciences from FCT-Portuguese Foundation for Science and Technology (UID/BIO/04565/2013 and UIDB/04565/2020), and from Programa Operacional Regional de Lisboa 2020 (Project Number 007317), as well as from project LISBOA-01-0145-FEDER-022231-the BioData.pt Research Infrastructure, is acknowledged. We acknowledge John Bennett, of the National Institute of Allergy and Infectious Diseases, NIH, Bethesda, USA, for kindly providing the L5U1 strain.

## Author contributions

M.C. conducted most of the experiments. D.P., C.L., P.P., M.O., and G.B. contributed to the RNA-seq data generation and analysis. R.S. contributed to the transcription factor screening. A.I.P. and A.M.F. contributed to the epithelium adhesion assays. R.V., A.T.N., M.O., and H.C. contributed to genetic manipulation and strain construction. C.F.D., E.N., and E.D. contributed to the AFM experiments. E.D. and M.C.T. designed and coordinated the full project. The manuscript was written by M.C., E.D., and M.C.T., with contributions of all authors. All authors have given approval to the final version of the manuscript.

## Competing interests

The authors declare no competing interests.
