## [Peer Review File · Communications Biology]

Reviewers' comments:

Reviewer #1 (Remarks to the Author):

The manuscript documents adherence of *C. glabrata* to polystyrene, silicone and PVC, with this interaction being essentially instantaneous. The study documents adherence to vaginal epithelial cells as well and shows that here as well there is essentially instantaneous with no substantial change in adherence characteristics over 60s.

Biofilm formation was assessed on polystyrene using a cell staining assay (Presto Blue). Deletion of two transcription factors – EFG1 and TEC1 reduces biofilm formation in this assay, and expression of these in a second strain increases biofilm formation. These genes seem to be involved in adherence to vaginal epithelia as well since deletion strains reduce adherence modestly from 20% to 15-17%. Expression in a second strain increases adherence modestly from 25% to perhaps 32 to 33%.

The authors carry out transcriptomics on planktonic cells versus biofilm cells and show gross changes in transcriptome. The Supplementary Tables showing changes are not included in the files for review. Notably, PWP5, AED2, and AWP13 are upregulated in biofilm and this upregulation depends on EFG1 and TEC1. Deletion of PWP5 has a reproducible albeit small effect on biofilm formation. Deletion of AED2 has a larger impact on formation of biofilm at 24h.

Interestingly, loss of EFG1 but not TEC1 reduces the initial adherence force to polystyrene and PVC. Loss of EFG1 also impacts adherence to epithelial cells.

Overall, the study is valuable. For completeness, there should be some analysis of PWP5 and AED2 mutants for adherence to surfaces, since the data begs the question of why PWP5 and AED2 mutants have these modest biofilm phenotypes – is it due to a primary adherence defect?

The standard in the field is that mutants need to be complemented to show that the phenotype is in fact due to the loss of the gene in question. The alternative is that experiments are repeated on independently generated mutants. I am not sure if this is feasible for all experiments but for some key conclusions, it is important to solidify the conclusion that the phenotype (eg adherence/biofilm formation) is due to loss of the EFG1 or TEC1. I don't think the RNAseq needs to be repeated on a second strain.

In general, the manuscript is quite descriptive. This is expected because of the complexity of the phenotypes being assessed, which likely integrate expression of many many genes. However, the authors could be more clear in putting their study in context. The most important thing is to indicate that these assays and the adherence being measured is likely highly variable between strains. Different strains of *C. glabrata* express different adhesins and therefore have different surface properties. The recent study of Valloteau shows that transcription factor changes dramatically impacts adherence to non-biologic surfaces. The data in this manuscript show that similar adherence effects are impacted by loss of different transcription factors. Martinez-Jimenez et al (PMID: 23392823 DOI:10.1007/s11046-013-9627-2_) have shown nicely that different strains of *C. glabrata* have different levels of adherence and are polymorphic for the major transcriptional regulator SIR3. Leiva-Peláez et al (<https://doi.org/10.1016/j.fgb.2018.05.005>) show conclusively that a highly hyperadherent clinical isolate has non-functional Sir3. Vale Silva showed that clinical isolates with differences in PDR1 also showed differences in adherence (Infect Immun. 2013 May;81(5):1709-20. doi: 10.1128/IAI.00074-13. Epub 2013 Mar 4.) These papers should be cited as they provide a clear demonstration that transcription factor differences between *C. glabrata* clinical strains result in gross changes in the kinds of adherence being measured in this manuscript. Basically, expression differences of surface components will obviously impact gross adherence measured at the level of the single cell, and the authors may wish to put their results in better context by acknowledging the impact of particular growth conditions chosen for these assays, and especially strain differences etc.

The standard in the field is that mutants need to be complemented to show that the phenotype is in fact due to the loss of the gene in question. The alternative is that experiments are repeated on

independently generated mutants. I am not sure if this is feasible for all experiments but for some key conclusions, it is important to solidify the conclusion that the phenotype is due to loss of the EFG1 or TEC1. I don't think the RNAseq needs to be repeated on a second strain.

Minor Points

Lines 185-189 are not clear. Please rephrase.

Figure 4b – is the extent of overexpression known for the plasmid constructs.

Figure 5 - These are tiny differences, changing adherence by less than 1.3 fold. Are the authors certain this is meaningful?

Y axis is shown as absorbance in multiple figures. This should be corrected.

For Figure 6, why is biofilm formation not assessed at 48h where the upregulation of PWP5 and AED2 is most dramatic?

The biofilm assay is a measure of total cell mass. Can the authors exclude a general effect of EFG1 or TEC1 loss on growth rate in the mutants that gives rise to the modest differences in biofilm amount. This would impact the claim that these are biofilm regulators as opposed to being generally required for growth.

Why are the PWP5, AED2 and AWP13 mutants not tested for impact on adherence to the various surfaces? This would help link the biofilm and adherence parts of the study.

Reviewer #2 (Remarks to the Author):

The manuscript by Cavalheiro et al. describes the use of single-cell force spectroscopy to study the adhesion of *Candida glabrata* to vaginal epithelial cells and several surfaces, in particular, those often found in indwelling devices. In addition, the authors explored the transcriptional network behind biofilm formation and found two putative transcription factors as key players in this process. Deletion of these genes affected biofilm formation, and gene transcription, in particular, those related to adhesion.

Overall, the study addresses an interesting subject, since no much information in currently known about transcriptional regulation during biofilm formation in *Candida glabrata*. However, there are some points that the authors should include to strengthen the study.

1.-During the analysis of adhesion, does the interaction require of the fungal metabolism? or only proteins or other components that are already in the wall? A control with killed cells (with UV, for example) could add relevant information to this point.

2.- No experimental proof is given to support the claim that the identified genes encode for transcriptional factors. The bioinformatics analysis is not enough. Moreover, there is no evidence to conclude these genes are the functional orthologs of those found in *C. albicans*.

3.- Figure 4. These results could be explained by defects in cell growth, and therefore not necessarily related to biofilm formation. The authors should explore whether the mutant cells have similar growth rates than parental control strains. There is a typo in the figure, it should be absorbance.

4.- The authors should explore whether the mutants lacking EFG1 or TEC1 have the ability to form extracellular matrix during conditions that promote biofilm formation.

5.- The lack of re-integrant control strains is notorious for this reviewer. The authors have to include these controls to really link the phenotypes with the disrupted genes.

6.- Please include the appropriate references to support the use of ACT1 to normalize data during expression analyses in *C. glabrata*.

Reviewers' comments:

Reviewer #1 (Remarks to the Author):

The manuscript documents adherence of *C. glabrata* to polystyrene, silicone and PVC, with this interaction being essentially instantaneous. The study documents adherence to vaginal epithelial cells as well and shows that here as well there is essentially instantaneous with no substantial change in adherence characteristics over 60s.

Biofilm formation was assessed on polystyrene using a cell staining assay (Presto Blue). Deletion of two transcription factors – EFG1 and TEC1 reduces biofilm formation in this assay, and expression of these in a second strain increases biofilm formation. These genes seem to be involved in adherence to vaginal epithelia as well since deletion strains reduce adherence modestly from 20% to 15-17%. Expression in a second strain increases adherence modestly from 25% to perhaps 32 to 33%.

The authors carry out transcriptomics on planktonic cells versus biofilm cells and show gross changes in transcriptome. The Supplementary Tables showing changes are not included in the files for review. Notably, PWP5, AED2, and AWP13 are upregulated in biofilm and this upregulation depends on EFG1 and TEC1. Deletion of PWP5 has a reproducible albeit small effect on biofilm formation. Deletion of AED2 has a larger impact on formation of biofilm at 24h.

Interestingly, loss of EFG1 but not TEC1 reduces the initial adherence force to polystyrene and PVC. Loss of EFG1 also impacts adherence to epithelial cells.

Overall, the study is valuable. For completeness, there should be some analysis of PWP5 and AED2 mutants for adherence to surfaces, since the data begs the question of why PWP5 and AED2 mutants have these modest biofilm phenotypes – is it due to a primary adherence defect?

Answer: Thank you for your very interesting question. We haven't test the *PWP5* and *AED2* mutants for adherence to the surfaces due to the results obtain on their expression. We have results showing that their expression is increased only at 24h but especially 48h of biofilm formation, which indicates that their role is more related to cell-to-cell adhesion and cohesion of the biofilm than a role in the first moments of adhesion. This is supported by the fact that at planktonic conditions, these adhesins are not controlled by Efg1 (Supplementary table 4). We tried to make this point clearer in the revised manuscript.

The standard in the field is that mutants need to be complemented to show that the phenotype is in fact due to the loss of the gene in question. The alternative is that experiments are repeated on independently generated mutants. I am not sure if this is feasible for all experiments but for some key conclusions, it is important to solidify the conclusion that the phenotype (eg adherence/biofilm formation) is due to loss of the EFG1 or TEC1. I don't think the RNAseq needs to be repeated on a second strain.

Answer: Indeed the reviewer is correct. As requested, we have tested the effect of complementing the deletion mutant strain with a plasmid expressing Efg1, performing the experiments of adhesion to abiotic and biotic surfaces by single-cell force spectroscopy. The phenotypes due to the loss of *EFG1* were fully confirmed, given that complemented strains recover the wild-type phenotype. These new results were included in the revised manuscript.

In general, the manuscript is quite descriptive. This is expected because of the complexity of the phenotypes being assessed, which likely integrate expression of many many genes. However, the authors could be more clear in putting their study in context. The most important thing is to indicate that these assays and the adherence being measured is likely highly variable between strains. Different strains of *C. glabrata* express different adhesins and therefore have different surface properties. The recent study of Valloteau shows that transcription factor changes dramatically impacts adherence to non-biologic surfaces. The data in this manuscript show that similar adherence effects are impacted by loss of different transcription factors. Martinez-Jimenez et al (PMID: 23392823 DOI:10.1007/s11046-013-9627-2_) have shown nicely that different

strains of *C. glabrata* have different levels of adherence and are polymorphic for the major transcriptional regulator SIR3. Leiva-Peláez et al (<https://doi.org/10.1016/j.fgb.2018.05.005>) show conclusively that a highly hyperadherent clinical isolate has non-functional Sir3. Vale Silva showed that clinical isolates with differences in PDR1 also showed differences in adherence (Infect Immun. 2013 May;81(5):1709-20. doi: 10.1128/IAI.00074-13. Epub 2013 Mar 4.) These papers should be cited as they provide a clear demonstration that transcription factor differences between *C. glabrata* clinical strains result in gross changes in the kinds of adherence being measured in this manuscript. Basically, expression differences of surface components will obviously impact gross adherence measured at the level of the single cell, and the authors may wish to put their results in better context by acknowledging the impact of particular growth conditions chosen for these assays, and especially strain differences etc.

Answer: Thank you for your suggestion. We have added this important discussion, together with the suggested references, in lines 510-513.

The standard in the field is that mutants need to be complemented to show that the phenotype is in fact due to the loss of the gene in question. The alternative is that experiments are repeated on independently generated mutants. I am not sure if this is feasible for all experiments but for some key conclusions, it is important to solidify the conclusion that the phenotype is due to loss of the *EFG1* or *TEC1*. I don't think the RNAseq needs to be repeated on a second strain.

Answer: As mentioned above, results with complemented strains have been performed and added to the previous results (See Figures 4, 7 and 9), solidifying that the observed phenotypes are indeed due to the loss of *EFG1* or *TEC1*.

Minor Points

Lines 185-189 are not clear. Please rephrase.

Answer: it has been changed.

Figure 4b – is the extent of overexpression known for the plasmid constructs.

Answer: Unfortunately, we don't know for sure. Whatever the case is, though, it is enough to recover, at least partially the wild-type phenotypes, when added to the corresponding deletion mutant strains.

Figure 5 - These are tiny differences, changing adherence by less than 1.3 fold. Are the authors certain this is meaningful?

Answer: Although the difference is relatively small, the results are statistically significant, based on multiple biological and technical replicates. We further believe that the impact in the overall phenotype is also clear.

Y axis is shown as absorbance in multiple figures. This should be corrected.

Answer: it has been corrected. Thank you for pointing it out.

For Figure 6, why is biofilm formation not assessed at 48h where the upregulation of PWP5 and AED2 is most dramatic?

Answer: *In vitro* *C. glabrata* biofilms are already at maturation phase at 24h, so at 48h we have more than mature biofilms. Given that our interest was to see the importance of these adhesins in the process biofilm formation, we've chosen 24h for the assessment of their significance.

The biofilm assay is a measure of total cell mass. Can the authors exclude a general effect of *EFG1* or *TEC1* loss on growth rate in the mutants that gives rise to the modest differences in biofilm amount. This would impact the claim that these are biofilm regulators as opposed to being generally required for growth.

Answer: We could not see any growth defect displayed by the deletion mutants, when compared to the wild-type parental strain. Representative growth curves are now displayed in Supplementary Fig. 3, and this issue is highlighted in the revised manuscript.

Why are the PWP5, AED2 and AWP13 mutants not tested for impact on adherence to the various surfaces? This would help link the biofilm and adherence parts of the study.

Answer: According to the results of expression of the RNA-seq data, these adhesins are not so relevant in the first seconds of adhesion of *C. glabrata*, not being activated by Efg1 in planktonic conditions. Instead, they seem to be relevant in latter stages of biofilm formation, probably in cell-to-cell adhesion. This results suggest that any test of the deletion mutants with SCFS on the medical-related surfaces would not produce differences from the wild-type. This notion is more clearly conveyed in the revised manuscript.

Reviewer #2 (Remarks to the Author):

The manuscript by Cavalheiro et al. describes the use of single-cell force spectroscopy to study the adhesion of *Candida glabrata* to vaginal epithelial cells and several surfaces, in particular, those often found in indwelling devices. In addition, the authors explored the transcriptional network behind biofilm formation and found two putative transcription factors as key players in this process. Deletion of these genes affected biofilm formation, and gene transcription, in particular, those related to adhesion.

Overall, the study addresses an interesting subject, since no much information in currently known about transcriptional regulation during biofilm formation in *Candida glabrata*. However, there are some points that the authors should include to strengthen the study.

1.-During the analysis of adhesion, does the interaction require of the fungal metabolism? or only proteins or other components that are already in the wall? A control with killed cells (with UV, for example) could add relevant information to this point.

Answer: That is an interesting question. However, given that the analysis with SCFS lasts only for 5s, we strongly believe that adhesion relies only on what is already present at the cell wall in the beginning of the assay. This notion was clarified in the manuscript.

2.- No experimental proof is given to support the claim that the identified genes encode for transcriptional factors. The bioinformatics analysis is not enough. Moreover, there is no evidence to conclude these genes are the functional orthologs of those found in *C. albicans*.

Answer: The *C. glabrata* Tec1 and Efg1 proteins described in this study are orthologs of two transcription factors, one each, in *C. albicans* (Tec1 and Efg1) and *S. cerevisiae* (Tec1 and Sok2), according to the *Candida* genome databases, based on the Yeast Genome Order Browser and the *Candida* Genome Order Browser databases. The *C. albicans* and *S. cerevisiae* TFs have been characterized and shown to bind directly to the promoters of their target genes. Indeed, according with the information deposited in the Yeastract+ database, CaTec, CaEfg1, ScTec and ScSok2 were proven to bind to 99, 675, 524 and 874 promoter regions, respectively. Altogether, although we understand the concern of the reviewer, we believe we have sufficient support to claim the role of TFs to the proteins under scrutiny in our study. This idea was more clearly put in the revised manuscript.

3.- Figure 4. These results could be explained by defects in cell growth, and therefore not necessarily related to biofilm formation. The authors should explore whether the mutant cells have similar growth rates than parental control strains. There is a typo in the figure, it should be absorbance.

Answer: We could not see any growth defect displayed by the deletion mutants, when compared to the wild-type parental strain. Representative growth curves are now displayed in Supplementary Fig. 3, and this issue is highlighted in the revised manuscript.

4.- The authors should explore whether the mutants lacking EFG1 or TEC1 have the ability to form extracellular matrix during conditions that promote biofilm formation.

Answer: Thank you for your important remark. The proposed evaluation was carried out and we show, in the revised manuscript, how the deletion of EFG1 or TEC1 genes affects the production of key EPS of the ECM (Supplementary Fig. 4), confirming that each regulator influences different aspects of ECM.

5.- The lack of re-integrant control strains is notorious for this reviewer. The authors have to include these controls to really link the phenotypes with the disrupted genes.

Answer: Results with re-integrant control strains have been performed and added to the previous results (See Figures 4, 7 and 9). Thank you for demanding this experiment.

6.- Please include the appropriate references to support the use of ACT1 to normalize data during expression analyses in *C. glabrata*.

Answer: A reference supporting this option was added to the revised manuscript.

REVIEWERS' COMMENTS:

Reviewer #1 (Remarks to the Author):

I find the revisions to be appropriate. The major technical issue (complementation) has been addressed. Some additional issues can be considered to be outside the scope of this paper. In general, this is a lovely piece of work which will be welcome in the field.

I suggest the following minor text change

However, when performing these comparisons, it is
512 important to keep in mind that differences in adhesion have been reported between different
C. glabrata
513 strains, related with the presence of different adhesins, regulation of the Sir Complex or the
Pdr1
transcription factor 52–54

change to:

However, when performing these comparisons, it is
512 important to keep in mind that differences in adhesion have been reported between different
C. glabrata
513 strains. Strain to strain variation can be related to the presence in the genome of different
adhesins, but also to differences in adhesin expression, which, for example, can be related to
strain differences in the function of the Sir Complex function or the Pdr1 transcription factor 52–54

Reviewer #2 (Remarks to the Author):

I thank the authors for the revised version of the manuscript, which addresses all my previous concerns. The study is indeed recommended for publication and I am sure it will be well received by the specialized community.

Reviewers' comments:

Reviewer #1 (Remarks to the Author):

I find the revisions to be appropriate. The major technical issue (complementation) has been addressed. Some additional issues can be considered to be outside the scope of this paper. In general, this is a lovely piece of work which will be welcome in the field.

ANSWER: Thank you for your positive evaluation of our manuscript.

I suggest the following minor text change

However, when performing these comparisons, it is important to keep in mind that differences in adhesion have been reported between different *C. glabrata* strains, related with the presence of different adhesins, regulation of the Sir Complex or the Pdr1 transcription factor

change to:

However, when performing these comparisons, it is important to keep in mind that differences in adhesion have been reported between different *C. glabrata* strains. Strain to strain variation can be related to the presence in the genome of different adhesins, but also to differences in adhesin expression, which, for example, can be related to strain differences in the function of the Sir Complex function or the Pdr1 transcription factor

ANSWER:

The text was corrected exactly as proposed.

Reviewer #2 (Remarks to the Author):

I thank the authors for the revised version of the manuscript, which addresses all my previous concerns. The study is indeed recommended for publication and I am sure it will be well received by the specialized community.

ANSWER: Thank you for your positive evaluation of our manuscript.